# ACC neural ensemble dynamics are structured by strategy prevalence

**Mikhail Proskurin[1,2], Maxim Manakov[1,2], Alla Karpova[1]***

[1]Janelia Research Campus, Howard Hughes Medical Institute, Ashburn, United States; [2]Department of Neuroscience, Johns Hopkins University Medical School, Baltimore, United States

**Abstract** Medial frontal cortical areas are thought to play a critical role in the brain's ability to flexibly deploy strategies that are effective in complex settings, yet the underlying circuit computations remain unclear. Here, by examining neural ensemble activity in male rats that sample different strategies in a self-guided search for latent task structure, we observe robust tracking during strategy execution of a summary statistic for that strategy in recent behavioral history by the anterior cingulate cortex (ACC), especially by an area homologous to primate area 32D. Using the simplest summary statistic – strategy prevalence in the last 20 choices – we find that its encoding in the ACC during strategy execution is wide-scale, independent of reward delivery, and persists through a substantial ensemble reorganization that accompanies changes in global context. We further demonstrate that the tracking of reward by the ACC ensemble is also strategy-specific, but that *reward* prevalence is insufficient to explain the observed activity modulation during strategy execution. Our findings argue that ACC ensemble dynamics is structured by a summary statistic of recent behavioral choices, raising the possibility that ACC plays a role in estimating – through statistical learning – which actions promote the occurrence of events in the environment.

## Editor's evaluation

This manuscript posits a novel role for the anterior cingulate cortex (ACC) in coding for sequential action strategies and the prevalence of each strategy. These findings provide important insight into ACC function and will therefore be of broad interest within the field of cognitive neuroscience. The evidence supporting the primary hypothesis is convincing.

*For correspondence:
alla@janelia.hhmi.org

Competing interest: The authors declare that no competing interests exist.

## Introduction

Flexibility in choosing one's behavioral strategy is a foundational characteristic of intelligent behavior, enabling rapid detection and adaptation to changes in the environment. In mammals, the brain's ability to adaptively change behavior is thought to depend on the coordinated action of medial frontal cortical areas that keep track of the information necessary for context-appropriate choices of strategy (*Domenech et al., 2020*; *Donoso et al., 2014*). Functional imaging studies in human subjects and non-human primates have suggested that the anterior cingulate cortex (ACC), in particular, is a critical cortical node that translates contextual information into strategy changes (*Domenech et al., 2020*; *Donoso et al., 2014*; *Hayden et al., 2011*; *Kolling et al., 2012*; *Sarafyazd and Jazayeri, 2019*; *Seo et al., 2014*). Less clear is what specific computations are instantiated in ACC neural ensemble dynamics, in part because analyses of the diverse neural responses in ACC have yielded few simplifying principles. Previous interpretations of the diversity of individual neural responses in frontal regions such as the ACC have included a direct role in enabling the separation of distinct contexts (*Rigotti et al., 2010*), and in tracking the animal's evolving motor state (*Musall et al., 2019*; *Stringer*

*et al., 2019*). However, unexpectedly specific ACC responses associated with seemingly self-directed strategic choices have also been observed in tasks that relaxed some of the experimental control over subjects' behavioral responses (*Schuck et al., 2015*; *White et al., 2019*), raising the possibility that more structured responses in the ACC remain to be discovered.

One hemodynamic study of the frontal cortical engagement in tasks with less external instruction highlighted the possibility that ACC is specifically engaged during self-guided exploration, with both the self-generation of a specific information-sampling strategy and its evaluation contributing to the observed signal (*Walton et al., 2004*). The advantage of choosing one's strategy for sampling information when learning in complex settings has long been appreciated (*Markant and Gureckis, 2014*); indeed, the idea that subjects learn more effectively when they self-direct their learning experience is central to many educational philosophies (*Boekaerts, 1997*; *Bruner, 1961*). Implicit in these notions is a key role for the self-guided component of knowledge acquisition. As such, it is notable that the medial frontal lobe, including the supplementary motor cortex (SMC) and the ACC, have been proposed to process information related to self, and to implement flexible, self-guided actions (reviewed in *Passingham et al., 2010*, but see *Schüür and Haggard, 2011*). The cingulate region also features prominently in emerging efforts to experimentally expose task-related information seeking and to uncover its neural underpinnings (*Wang and Hayden, 2020*; *White et al., 2019*). Thus, evaluating neural activity in behavioral settings that require self-directed discovery of effective strategies may represent a fruitful direction in the search for organizing principles of frontal cortical neural ensemble dynamics.

In this study, we incorporate self-guided search for task structure into the experimental design by requiring rats – in an apparatus that has 'left' and 'right' nose ports – to discover (without any explicit instruction) a specific rewarded sequence of 'Left'/'Right' choices from a larger set of structured possibilities. During behavioral sessions that incorporate blocks where the latent target sequence remains the same over 200–500 trials, rats infer – in each block – the specific target sequence through self-guided exploration, strongly biasing their choices accordingly within tens of trials following unsignalled block transitions. Nevertheless, throughout each block, rats also continue to periodically sample alternative sequences, favoring those previously experienced as other latent targets. As such, each sequence in the task set of structured possibilities gets executed in several discrete global contexts – which we define as inferred target sequence – as well as in a constantly changing local context – which we define as the specific set of self-guided sequence choices in recent past. We demonstrate that under these conditions, ensemble activity in ACC during the execution of a specific sequence tracks both global and local contexts in which that sequence is being executed. In particular, a summary statistic that we capture as the prevalence of that sequence in the last 20 trials, can be reliably decoded from the ACC ensemble activity throughout the extent of sequence execution. Remarkably, the tracking during sequence execution of the local sequence prevalence persists even as ACC activity is markedly re-organized between global contexts. We further demonstrate that reward is also tracked in the ACC in relation to specific discovered sequences but that *reward* tracking alone is insufficient to explain the observed prevalence encoding during sequence execution. Our findings demonstrate that the ACC continuously represents a summary statistic reflecting the local prevalence of self-guided behavioral strategies, suggesting that such encoding of recent self-action might be a fundamental part of the animals' algorithmic approach to discovering adaptive strategies in complex settings.

## Results

### Behavioral task for self-guided search for task structure and strategy encoding in ACC

To examine frontal cortical neural dynamics during self-guided exploration of a complex environment, we developed a behavioral task that requires rats to discover, without any explicit instruction, a specific sequence of 'Left'/'Right' choices that is preferentially rewarded. A typical behavioral session in this framework comprises a series (750–2000) of self-initiated trials that involve a choice between two options – left or right nose port – with a reward being delivered at the choice port upon execution of the latent (i.e. not explicitly revealed) target sequence of choices – such as right-right-left, 'RRL'. Our behavioral paradigm requires center-port entries between each choice of a side port. Hence, execution of the length 3 sequence 'RRL' requires the series of 6 port entries: cRcRcL (*Figure 1a*). The

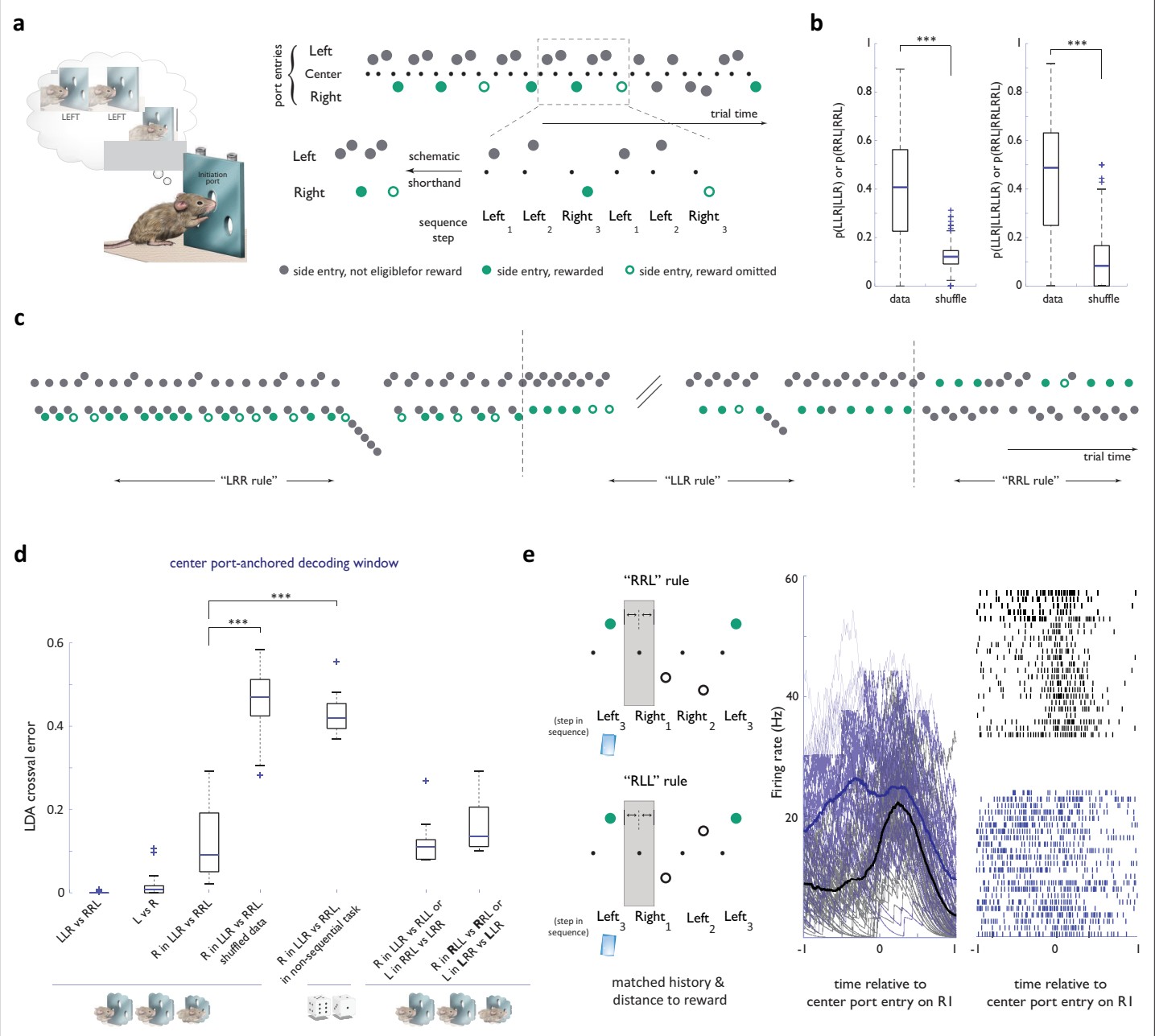

**Figure 1.** Strategy encoding in the ACC. (**a**) Left panel: Concept of the behavioral task. After initiating at the center port, the animal is eligible to receive a reward only if his sequence of past choices conforms to a latent target sequence, like 'Left-Left-Right'. Note that the identity of the latent target is not otherwise cued in any way. Liquid reward was delivered directly at choice ports. Right panel: Schematic of the notation used for behavioral data presentation. Note that nose port entries are omitted from schematics in other panels for simplification. (**b**) Probability of target sequence concatenations across the behavioral dataset. Shuffle randomized trial numbers. (**c**) Sample behavioral trace, in trial time, around two block transitions. (**d**) Cross-validated performance errors for linear classifiers trained to distinguish components of different strategies based on ACC neural activity in a decoding window anchored on center port initiation entry. n=36 sessions, N=4 animals for all basic sequence task comparisons; n=9 sessions, N=3 animals for competitor sessions; n=11 sessions, N=1 animal for circularly permuted strategies; n=8 sessions, N=3 animals for 1st R in 'RRL' vs 'RLL' (or L in 'LLR' vs 'LRR' decoding). See legend in *Figure 1—figure supplement 3* for more details on the non-sequential task (**e**) Activity traces for an example ACC neuron in the 2 second window around center port entry on the R1 step of 'RLL' and 'RRL' sequences -a step matched both in immediate history and distance to reward. Black traces: 'RLL' rule; blue traces: 'RRL' rule. ***, p<0.001.

The online version of this article includes the following figure supplement(s) for figure 1:

**Figure supplement 1.** Robustness of sequencing behavior.

**Figure supplement 2.** Deviations from the dominant sequence contain explorations of previously reinforced alternatives.

*Figure 1 continued on next page*

*Figure 1 continued*

**Figure supplement 3.** Strategy encoding in the ACC.

requirement for center port entries between each side port entry was chosen in part to control the movements associated with the execution of a given sequence; with this requirement, the rat's movements to select each instance of 'R' in the sequences 'RRL', 'LLR', etc., are behaviorally constrained to involve withdrawing from the center port, then moving to and entering the right port. This center port requirement, therefore, facilitates the separation of neural signals specifically associated with the sequence of 'R' and 'L' choices from the movements associated with selecting them (see below).

To ensure that animals continue to self-direct a search for the relevant task structure throughout each behavioral session, we changed the latent target sequence identity in an unsignalled manner every 250–500 trials (Methods). To collect a sizable amount of reward in this task, an animal needs to discover the target sequence and locally structure its choices to preferentially conform to the target sequence, yet remain flexible enough to efficiently adapt to unsignalled changes in its identity (*Figure 1a–b*). For most experiments in this study, we restricted the set of possible latent target sequences in any individual session to some, or all, of the four non-trivial three-step sequences ('LLR', 'RRL', 'LRR', and 'RLL'). Under these conditions, expert animals flexibly and reliably discovered and exploited the locally-relevant latent target: within tens of trials following block transitions, the animals' choices typically became dominated by the new target sequence, even in the presence of substantial (up to 30%) sporadic omission of reward for the properly executed target sequence (*Figures 1b–c and 2b*, *Figure 1—figure supplement 1*).

The observed behavioral flexibility did not result from an inability to commit to the discovered target sequence. Indeed, the target sequence clearly dominated behavioral choice streams during blocks of stability, with long concatenations of the target sequence frequently evident in the behavioral records of expert animals (*Figures 1b–c and 2b*, *Figure 1—figure supplement 1b*). Nevertheless, clear deviations from this dominant pattern, with the animals' choices appearing instead to conform to other possible target sequences, were also present within all blocks (*Figures 1c and 2a–c*). Although some of the deviations from the currently rewarded target sequence may represent errors of execution, the presence of bouts that often contain direct concatenations of previously reinforced sequences (*Figure 1—figure supplement 2a–d*) argue that at least some of these deviations represent transient exploratory resurgence of alternative sequences. Furthermore, while such transient deviations away from the dominant sequence were significantly more likely to follow the absence of an expected reward (*Figure 1—figure supplement 2i*), similar strategy deviations were present when no reward was omitted (*Figure 1—figure supplement 2f–h*), suggesting that animals continue to sporadically sample other sequences even when not extrinsically prompted. Thus, with only two well-defined individual actions ('left' and 'right'), this framework provides a means to engineer a rich and flexible repertoire of multi-step sequences that animals evaluate in various behavioral contexts. As such, this setting presented us with an opportunity to evaluate whether the richness of contextual variation that accompanies the added agency of self-guided strategy selection is reflected in the ACC ensemble dynamics. Indeed, the ACC is thought to play a central role in motivating extended, multi-step behaviors (*Holroyd and Yeung, 2012*), and our previous work demonstrated the rodent ACC homologue's involvement in unguided discovery of action sequences (*Tervo et al., 2021*). Nevertheless, before examining any putative encoding of contextual variation, we first verified that the specific choice of sequence in this more complex sequence task is reflected in the ACC neural dynamics by establishing that individual sequences could be decoded from ensemble activity. For these sequence decoding analyses, we focused on sequence instances that both matched the latent rewarded target and were executed after that sequence had established dominance in the animal's choices.

A simple linear classifier could reliably distinguish ACC activity between 'LLR' and 'RRL' sequences – the two main target sequences in our dataset – when provided with firing rates of all active cells in 500 ms windows centered on side port entries for the three steps in each sequence (*Figure 1—figure supplement 3* median cross-validated classification error 0.0, IQR 0.006, Methods). Robust classification was also achieved when the decoding window for each of the three sequence steps was instead anchored on the initiation (center) port entry common to all trials, prior to when the overt choice on that trial was made by the animal (*Figure 1d*, median cross-validated classification error 0.0, IQR 0.007, Methods). Nevertheless, any interpretation of the decodability of the full 'LLR'

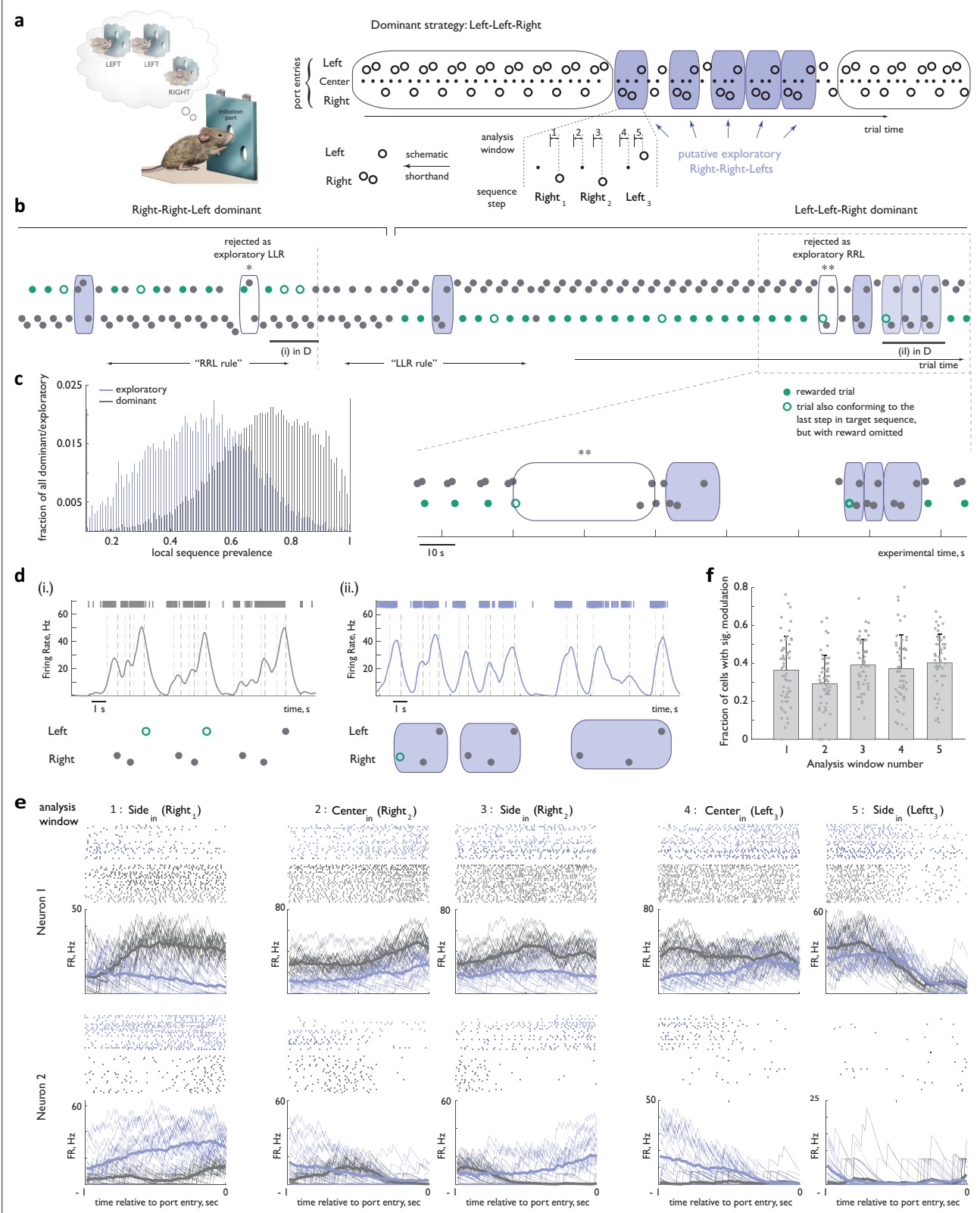

**Figure 2.** Activity of ACC neurons associated with a specific sequence of actions changes depending on whether that sequence represents the dominant strategy or is a transiently re-explored alternative. (**a**) Left panel: Concept of the behavioral task (same as in *Figure 1*). Right panel: (Top) Block-wise structure promoted the local pursuit of one dominant sequence (here, 'Left-Left-Right', as shown in the thought bubble) at the expense of others. The dominant strategy was occasionally interrupted by explorations of alternative sequences (here, likely 'Right-Right-Left', blue shading).

*Figure 2 continued on next page*

*Figure 2 continued*

(Bottom) Five analysis windows chosen to minimize trajectory confounds were anchored on center- and side-noseport entry events associated with each of the three steps in the sequence, with the exception of the first center port entry. The latter omission was chosen to minimize the contribution from feedback-related activity modulation associated with preceding choices. (**b**) Top panel: Sample behavioral trace, in trial time, around a block transition. Bottom panel: boxed region of the behavioral trace, in trial time. Note a marked preference at the beginning of the behavioral trace for 'Right-Right-Left' and at the end for 'Left-Left-Right'. Putative exploratory sequences are shaded blue (see Methods for details about how exploratory sequences were identified). *, rejected as an exploratory sequence because it overlapped with the dominant sequence to its left, and did not have the temporal profile consistent with sequence marking (see text) to be rescued from the 'discard' group; **, rejected as a putative exploratory sequence because of an overlap with the dominant sequence to its left, and an unusually long break between putative steps 1 and 2. (**c**) Distribution of local sequence prevalence values for 'Left-Left-Right' and 'Right-Right-Left' across all dominant and exploratory instances in implanted animals. (**d**) Activity of an example ACC neuron for three concatenated unrewarded instances of 'Right-Right-Left' at the end of the dominant epoch in (**b**) (left panel) and during an exploratory bout following subsequent dominance of 'Left-Left-Right' (right panel). Dashed lines indicate beam breaks at port entries. Bars at the top of the panel: raw spike train. (**e**) Activity of two other ACC neurons from the behavioral session in (**b**), aligned in the five one-second analysis windows anchored on port entry events. Grey: dominant instances of 'Right-Right-Left' that followed an unrewarded 'Right-Right-Left'. Blue: exploratory instances of 'Right-Right-Left'. (**f**) Fraction of all recorded ACC units that displayed a significant modulation between 'dominant' and 'exploratory' contexts for each of the five analysis windows. Individual points correspond to different sessions. Error bars represent standard deviation. n=35 sessions, N=4 animals.

vs 'RRL' ACC representations is confounded by a strong difference in the encoding of the individual steps composing the sequences ('L' vs 'R'; *Figure 1d*, *Figure 1—figure supplement 3*, median cross-validated classification error 0.008, IQR 0.016 and 0.009, 0.0017 for side port-centered and center port-centered decoding windows respectively). To address this confound, we tested whether the encoding of the same action, such as 'R', differs between 'LLR' and 'RRL' sequences. We indeed found that even controlling for the specific action in this manner, the corresponding neural activity could be used to accurately decode whether that action was embedded in 'LLR' versus 'RRL' sequences (*Figure 1d*, *Figure 1—figure supplement 3*; median cross-validated error 0.10, IQR 0.14 and 0.08, 0.13 for side port-centered and center-port centered decoding windows, respectively). Moreover, when we performed an equivalent analysis for all sets of consecutive 'Left', 'Left', 'Right' and 'Right', 'Right', Left' triplets found in the choice streams generated when animals made largely independent choices on individual trials (see a detailed explanation in the legend for *Figure 1—figure supplement 3*), we found a significantly higher classification error (*Figure 1d*, *Figure 1—figure supplement 3*; median cross-validated error 0.42 and IQR 0.05 for the non-sequential setting vs 0.10, IQR 0.14 for the sequence task, $p < 10^{-12}$, Wilcoxon rank-sum test), suggesting that the differential encoding of 'R' in 'LLR' vs 'RRL' reflects some aspect of the task structure.

The observed differences in ACC representations for a given 'L' (or 'R') action within the sequences 'LLR' and 'RRL' could reflect the differential encoding of distinct strategies that are currently being pursued. However, the 'L's (or 'R's) in these different sequences are also associated with differences in the immediate history of other actions ('L' in 'RRL' follows an 'R', while one of the 'L's in 'LLR' follows an 'L') and in their proximity of rewards (no 'R' in 'RRL' is ever rewarded, whereas most 'R's in 'LLR' are). We therefore carried out additional analyses to ask whether either of these factors – surrounding actions, or reward proximity – could account for the apparent strategy encoding in ACC, taking advantage of a subset of sessions that tasked our animals with discovering additional three-letter target sequences.

To test for the influence of surrounding actions on sequence encoding, we examined behavioral epochs when distinct latent targets were associated with matching (but permuted) sequences of actions leading rats to produce matched behavioral streams despite pursuing different target sequences. For example, rats that discovered the latent target 'RRL' often evidenced this by producing long concatenations of the correct sequence ('…RRLRRLRRL…'). When these same rats discovered the different target 'LRR' and adopted the appropriate and distinct dominant strategy of repeating the 'LRR' sequence, the resulting behavioral stream matched the one previously observed with the 'RRL' target ('…(L)RRLRRLRRL(RR)…'). In such cases, we found that the currently dominant target sequence still could be readily decoded from ACC activity during individual actions such as 'L' despite their being embedded in indistinguishable choice streams (e.g. '…RRLRR…'; *Figure 1d*, *Figure 1—figure supplement 3*; median cross-validated classification error 0.10, IQR 0.04 and 0.11, 0.05 for side port-centered and center port-centered decoding windows, respectively).

We adopted a similar approach to examine whether the ability to decode the currently dominant strategy persisted when controlling for both the immediate past choice and for proximity to reward.

For example, we assessed how well the activity during the first 'L' could be used to decode a dominant strategy of 'LRR' versus 'LLR'. In this case, during concatenations of the currently rewarded sequences, the proximity from the first 'L' to reward delivery is matched (as is the immediately preceding action). Here too, despite controlling for reward proximity, the currently dominant strategy could readily be decoded (*Figure 1d*, *Figure 1—figure supplement 3*; median cross-validated classification error 0.13, IQR 0.09 and 0.14, 0.11 for side port-centered and center port-centered decoding windows respectively). Combined, these observations argue that individual multi-step sequential strategies have distinct ACC representations, permitting us to next evaluate whether the ACC representation of any specific sequence is further modulated by the specific context in which a particular instance of that sequence is being executed.

## Identification of distinct sequence strategies in the behavioral stream

To set the stage for examining contextual modulation of ACC's sequence representation, we first sought to identify all instances of individual sequence targets in the task set. The robust performance our rats displayed on this task (*Figures 1b and 2c*, *Figure 1—figure supplement 1*) suggests that tasks requiring discovery of latent structure through self-guided exploration align particularly well with how brains naturally make sense of complex environments (*Gottlieb and Oudeyer, 2018*; *Tervo et al., 2016*; *Wang and Hayden, 2021*). However, such an experimental framework poses challenges for parsing behavioral stream to identify specific sequence instances evaluated by the animal. Parsing a continuous stream of left and right choices to identify 'legitimate' sequence instances is easy for the target sequence once it has been discovered and has locally established dominance in the animal's choices. Indeed, a high prevalence of target sequence concatenation under such 'dominant' condition, and a scarcity of choices that conform to other patterns, argue that almost every instance of a pattern conforming to the latent target is likely to be an instance of that sequence actually evaluated by the animal (see Methods for a detailed description of the filter used to select 'dominant' sequence instances). In contrast, parsing is much harder for the deviations from the dominant target outside of clear sequence concatenations. This is particularly challenging in cases where the task set contains circularly permuted strategies: does a 'LRRL' deviation during a 'LLR' block reflect a 'LRR' instance, a 'RRL' instance, or neither? While the additional uncertainty inherent in parsing circularly permuted sequences can be resolved by focusing on sessions that contained only 'LLR' and 'RRL' blocks, not even every apposition of 'Left', 'Left', and 'Right' choices in a 'RRL' block will reflect an exploratory instance of the 'LLR' sequence. We therefore next sought to delineate an objective criterion for including any lone putative exploratory sequence instance in the subsequent investigation; with the exception of a few pre-specified control experiments, the remaining analyses are restricted to 'LLR'/'RRL' block switches.

Our approach was grounded in the expectation that animals would pause, if only briefly, at the side nose port on the last step of a true exploratory sequence instance, marking sequence completion. Under this assumption, putative exploratory sequence instances for which the duration of the third step exceeds a preset threshold, can be objectively included in the 'exploratory' dataset (i.e. the set of instances for the sequence of interest executed in the global context of another inferred latent target). Indeed, clear shifts to longer within-side-port and side-to-side durations for step three – as compared to steps one and two – were present across all 'dominant' sequence instances for which the otherwise scheduled reward was omitted (*Figure 1—figure supplement 2*, Methods), permitting an unbiased selection of the specific threshold. The 'exploratory' dataset thus included sequence concatenations and lone sequence instances that passed the temporal selection threshold (see Methods for the full description of the selection procedure). Overall, the rich contextual variation associated with the execution of different instances of any specific target sequence in both 'dominant' and 'exploratory' conditions provided us with an opportunity to determine whether examining frontal cortical neural activity through the lens of this natural contextual variation might reveal dynamics that would shed additional light on the computations performed by these circuits.

# Functional reorganization of the ACC network constrains its representation of a specific strategy in distinct inferred global task contexts

ACC is thought to guide contextually-appropriate strategy selection (reviewed in *Heilbronner and Hayden, 2016*; *Holroyd and Verguts, 2021*; *Kolling et al., 2016*; *Monosov et al., 2020*; *Shenhav et al., 2016*), prompting us to begin our analysis of task-related neural responses by evaluating how *global* behavioral context is reflected in ACC ensemble dynamics; we define the global context by the sequence that locally dominates the animal's choices – likely a manifestation of the animal's inference about the currently relevant task structure. Specifically, we first sought to determine if ACC activity associated with a specific behavioral sequence is reorganized depending on whether that sequence represents a dominant behavioral strategy or is sampled as a part of an exploratory bout, without classifying the latter global context further according to the identity of the inferred target sequence. Indeed, while abrupt changes in the activity of ACC neurons coincident with behavioral transitions to exploration have been reported previously (*Durstewitz et al., 2010*; *Emberly and Seamans Jeremy, 2019*; *Karlsson et al., 2012*; *Powell and Redish, 2016*), whether these transitions in ACC activity merely mark the behavioral state change, or reflect an actual task-related re-organization of activity whereby representations of individual strategies are also marked with contextual content, remains unclear. Targeted recordings of ACC activity – performed in a wireless configuration that did not impair the animals' behavioral flexibility – revealed that marked differences in activity associated with the execution of specific sequences in 'dominant' versus 'exploratory' contexts could indeed be readily observed across many ACC neurons, frequently evident even in non-trial averaged activity traces (*Figure 2d*). To facilitate detailed comparisons of neural activity in the face of inevitable variability in the spatio-temporal profiles of movements between nose ports for different instances of sequence execution, we have focused all analyses on five 500 millisecond windows anchored on center and side noseport entry events associated with the individual steps in the sequence (*Figure 2a*, see also a more detailed evaluation of potential motor confounds below).

An alignment of sequence-related activity in the five constrained analysis windows across different sequence instances exposed various forms of contextual activity modulation in individual cells (see *Figure 2e* for alignment of spike rasters, *Figure 3b* for alignment of heatmap representations). Indeed, both decreases, increases, and mixed modulations of activity in the exploratory context relative to that observed when the sequence represented the dominant strategy were observed (see *Figures 2e and 3a* for examples). At least 76% of all recorded units (787 of 1042 total ACC recorded units, with no separation of potential interneurons) displayed significant modulation in at least one of the five analysis windows – a lower bound given the limitation that the modest size of the exploratory dataset places on the statistical power of these analyses – and a roughly equal fraction of all units displayed modulation in each of the analysis windows (*Figure 2f*). Thus, activity changes related to the inferred global task context are present at the single-cell level in many ACC neurons.

To examine these contextual changes in the ACC representation associated with a specific behavioral sequence in more detail, we next evaluated population responses in a neural state space. For these analyses of potential contextual re-configurations, we used a state-space framework that assigns the mean firing rate of an individual neuron in each of the five analysis windows to a single dimension. Consequently, a point in this five-dimensional state captures the activity of that neuron during one instance of sequence execution (*Figure 3a–b*, Methods). The ensemble representation, in turn, is captured by further expanding the state space dimensionality to include these five dimensions for each of the simultaneously recorded neurons (*Figure 3c*). When we used similarity matrices to visualize instance-to-instance distance between ACC representations for a specific sequence in the full state space, a substantial separation of the two clusters formed by the 'dominant' and 'exploratory' instances of sequence execution was readily evident (*Figure 3d*). Indeed, the Euclidean distance between the centroids of these experimentally observed 'dominant' and 'exploratory' groups differed markedly from that observed when the 'dominant' and 'exploratory' labels were randomly shuffled across the set of all instances of sequence execution (*Figure 3d*, 0.73±0.18 for experimental data vs 0.268±0.077 for shuffled data, $p<10^{-12}$, Wilcoxon rank-sum test, n=35 sessions, N=4 animals; 2.6+/- 1.9 block changes per session). Moreover, the mean Euclidean distance in the activity state space between two 'dominant' blocks separated in time was significantly smaller than the mean pairwise distances between a dominant and exploratory block, arguing against the possibility that the observed

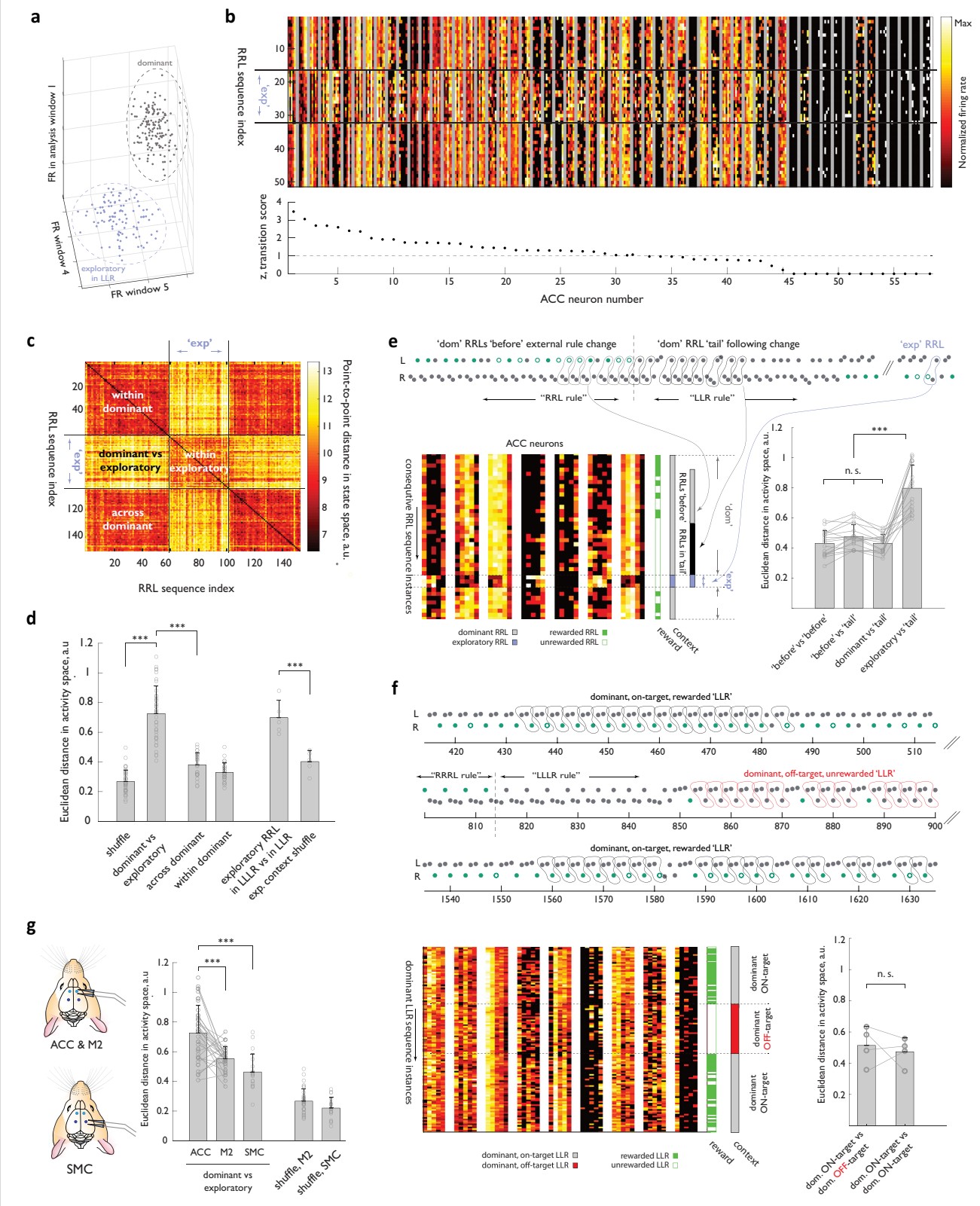

**Figure 3.** Representational transitions in ACC reflect large-scale functional reorganizations of the ACC network between inferred global behavioral contexts. (**a**) Schematic of the activity state space for an individual neuron. Three of the five dimensions corresponding to the analysis windows are shown. Two clouds schematize activity of that neuron associated with a specific behavioral sequence for all dominant (grey) and exploratory (blue) instances of the sequence. (**b**) Heat map representations of normalized activity associated with 'Right-Right-Left' sequence execution of 58

*Figure 3 continued on next page*

*Figure 3 continued*

simultaneously recorded ACC neurons. Different sequence instances are stacked vertically, with two 'dominant' blocks separated by a period when the 'Right-Right-Left' sequence was occasionally explored in the background of 'Left-Left-Right' dominance. 'exp', 'exploratory' instances. Neurons are arranged according to a 'transition score' defined as the distance between the two cloud centroids normalized by root mean of variance within each cloud (see Methods). (**c**) Similarity matrix for an example session, comparing the ACC ensemble activity across 'Right-Right-Left' instance pairs, using Euclidean distance in the network state space. Black lines indicate the boundary between 'dominant' and 'exploratory' contexts. (**d**) Euclidean distance (RRL instance-to-instance) in the state space between the relevant 'dominant' and 'exploratory' clouds for the experimental ACC data, and for the control state space, where the labels of 'dominant' and 'exploratory' (or of the specific 'exploratory' context) were randomly shuffled across the dataset. n=35 sessions, N=4 ACC-implanted animals. (**e**) Behavior of the ACC ensemble during persistence of the dominant strategy past the unsignalled transition in the rewarded target.(**Top panel**) Example behavioral transition with a long dominant RRL 'tail'. (**Bottom left panel**) Heat-map representation of the activity of 5 ACC cells for consecutive RRL instances before (i.e. when RRL dominance coincided with it being the rewarded target, 'before' in panel), during (when RRL continued to dominate the animal's choices but the target sequence had changed, 'tail' in panel) and after the behavioral transition. Note that the later set of examples included several exploratory instances (executed much later, once another sequence had established dominance, 'exp' in the panel), as well as several of the subsequent instances of 'dominant' RRL later in the session. 'exp', 'exploratory; dom', 'dominant'. (**Bottom right panel**) RRL instance-to-instance Euclidean distance in the state space across these distinct epochs for all long 'dominant tail' examples. n=12 sessions, N=4 animals. (**f**) Behavior of the ACC ensemble during 'ON-target' and 'OFF-target' persistence with a dominant 'LLR' sequence. (**Top panel**) Example behavioral trace from two 'ON-target' epochs and one 'OFF-target' epoch within the same behavioral session. Note that the animal had responded to a block change and adjusted his strategy before settling into an 'OFF-target' persistence with 'LLR' (middle epoch). (**Bottom left panel**) Heat-map representation of the activity of nine ACC cells for LLR instances in the three epochs. (**Bottom right panel**) Instance-to-instance Euclidean distance in the state space between an 'ON' and 'OFF' contexts and between two 'ON' contexts. n=4 sessions, N=2 animals. (**g**) Euclidean distance between the centroids of the 'dominant' and 'exploratory' clouds for the experimental ACC, M2, and SMC data, and for the control state spaces, where the labels of 'dominant' and 'exploratory' were randomly shuffled across the dataset. n=37 sessions, N=4 ACC/M2 -implanted animals; n=18 sessions, N=3 SMC-implanted animals. Error bars represent standard deviation. n.s., not significant, ***, p<0.001.

The online version of this article includes the following figure supplement(s) for figure 3:

**Figure supplement 1.** Behavior at block transitions is not a function of reinforcement.

**Figure supplement 2.** Representational transitions are less prominent in the caudal portion of the medial frontal lobe.

representational transitions in ACC arose from an instability in neural recordings (*Figure 3c and d*). Thus, ACC ensemble activity markedly changes its representation of a particular behavioral sequence when that sequence no longer represents the locally dominant behavioral strategy but settles back into a similar state every time the animal returns to its pursuit of that sequence over all others.

The strong similarity of ACC ensemble configurations across distinct, temporally segregated 'dominant' contexts (*Figure 3e*) argues that these representational rearrangements reflect something other than a mere episodic record of distinct behavioral contexts. Given that ACC is thought to encode signals that convey the value of pursuing alternative courses of action as opposed to the current, default action plan (*Behrens et al., 2007*; *Blanchard and Hayden, 2014*; *Hayden et al., 2009*; *Karlsson et al., 2012*; *Kolling et al., 2012*; *Kolling et al., 2014*; *Ma et al., 2016*; *McGuire et al., 2014*; *O'Reilly et al., 2013*; *Powell and Redish, 2016*; *Procyk et al., 2000*; *Schuck et al., 2015*; *Tervo et al., 2021*), the abrupt transitions in the ACC representation of specific sequences we observed may be designed to tag these representations simply as 'dominant'/'default' or 'exploratory'/ 'alternative'. However, broader contextual signals are also thought to be present in this cortical region (*Caracheo et al., 2018*; *Euston et al., 2012*; *Seamans and Floresco, 2022*; *Shenhav et al., 2016*; *Tomlin et al., 2006*), and thus the ACC representational transitions may carry a richer contextual content. To distinguish these possibilities, we compared, in a small subset of sessions, the ACC representation of exploratory 'RRL's in two separate contexts: the 'LLR' context (i.e. one where the animal otherwise pursued 'LLR' as the default strategy) and the 'LLLR' context. Despite the limited statistical power of these smaller exploratory datasets, representational changes between these distinct exploratory contexts were clearly evident both at the single neuron and the population level (*Figure 3e*, Euclidean distance of 0.70+/-0.11 for two sets of exploratory RRL representations vs 0.40+/-0.08 for a control set with scrambled contextual labels, p<0.008, Wilcoxon rank-sum test, n=5 sessions, 2 animals). Notably, the marked restructuring of ACC network configuration that underpins the differential encoding of exploratory 'RRL's in these two contexts argues against the possibility that these large scale-reconfigurations merely reflect the absence or presence of reward for a given sequence in that task epoch since none of the 'RRL's in *either* context were ever rewarded. Rather, these observations support the notion that the ACC network functionally reconfigures in distinct behavioral contexts and

argues that the inferred global contextual content of representational transitions in ACC is richer than the 'dominant'/'exploratory' dichotomy.

Our parsing of the global behavioral context based on the dominant strategy chosen by the animals rather than based on the experimentally imposed identity of the target sequence is in keeping with the view that the medial prefrontal cortex does not simply track external cues that situate the animal in time and place, but instead reflects the state of the animal's emotions and beliefs (*Caracheo et al., 2018*; *Euston et al., 2012*; *Seamans and Floresco, 2022*). Nevertheless, the externally imposed task context (i.e. the identity of the latent target sequence at any moment) and its parsing by the animal that presumably shapes the choice of the dominant strategy are strongly correlated in expert animals (*Figure 1—figure supplement 1*). Thus, to further probe the validity of parsing based on the animal's behavior, we next examined more closely the ACC neural ensemble activity evolution in two specific cases where the dominant strategy differed from the target sequence. First, at block transitions between the familiar 'LLR' and 'RRL' target sequences, the dominance of the previously rewarded target often transiently persisted in face of a latent rule change (*Figure 3e*). Ensemble states associated with sequence execution during such dominant sequence 'tails' did not cluster away from states observed earlier in the 'dominant' context, and were equally distant from the exploratory cluster (*Figure 3e*, Euclidean distance of 0.43+/-0.06 for sequence instances within 'tails' to other dominant sequence instances vs 0.80+/-0.15 for sequence instances within 'tails' to exploratory sequence instances, $p<10^{-6}$, Wilcoxon rank-sum test, n=17 'tail' examples, N=12 sessions, N=4 rats, 2.1+/-1.1 block changes per session). Second, in sessions where the target sequence set included not only the familiar 'LLR' and 'RRL' but also the recently introduced 'LLLR' and 'RRRL', animals' choices in the 'LLLR' and 'RRRL' blocks were often dominated by the cognate familiar shorter sequence due to a lack of proficiency with the longer targets. In such cases, ensemble states associated with such 'dominant' but unrewarded 'LLR' sequence instances also did not cluster far away from ones observed earlier in the same session when the dominant 'LLR' strategy matched the rewarded target (*Figure 3f*, Euclidean distance of 0.47+/-0.10 for across ON-target dominant contexts vs 0.53+/-0.12 between OFF-target and ON-target dominant contexts, p=0.68, Wilcoxon rank-sum test, n=4 sessions, N=2 animals). Furthermore, since persistence with an unrewarded sequence in this case occurred after the animals had detected an unsignalled block change and switched away from the previous strategy (*Figure 3f*), the similarity of ensemble states cannot be explained by a lack of rule switch awareness that potentially confounds the dominant 'tails' example. Together with our previous observation that the ACC neural representation associated with a specific probabilistically rewarded action differed markedly depending on the strategy-related context (*Tervo et al., 2021*), the unambiguous co-segregation of the ensemble representation for dominant sequence instances executed in no reward condition we observed here further argues against the simple 'reward'/'no reward' explanation of network reconfigurations. More broadly, these observations support parsing of the behavioral context through the lens of the selected strategy.

We also investigated whether the contextual reorganization of ensemble activity we observed in the ACC reflects an ACC-specific computation or is present in other parts of the frontal cortex. Indeed, neural correlates of decision task parameters have long been observed across many frontal cortical areas (*Cisek, 2012*; *Cisek and Kalaska, 2010*; *Hunt and Hayden, 2017*; *Yoo and Hayden, 2018*), and recent advances in the ability to simultaneously record from tens of thousands of neurons across many brain areas in the mouse have only further emphasized how widely distributed the task-related encoding can be in the brain (*Steinmetz et al., 2019*; *Stringer et al., 2019*). We focused on two distinct regions along the rostro-caudal axis of the medial lobe. At the rostral end, we targeted, within the same set of recording sessions, the immediately dorsal to the ACC part of the premotor cortex (henceforth, M2) that contains, among other areas, the putative rat homologue of the frontal orienting fields recently implicated in a range of sensory-guided decisions (*Ebbesen et al., 2018*). At the caudal end, we probed, in a separate set of animals, the neural dynamics in a premotor region that shares key afferent/efferent projection patterns with the primate SMC and is functionally required for self-guided action sequencing in our behavioral framework (*Figure 3—figure supplement 2*, Methods and (Manakov, Proskurin et al, *in preparation*); henceforth, SMC). Analysis of the M2 and the SMC ensemble representations associated with individual behavioral sequences over the course of long behavioral sessions revealed that although representational changes could also be observed in those parts of the medial lobe, these changes were significantly less pronounced and affected a

smaller fraction of the recorded units, especially in the more caudal SMC (*Figure 3g*, *Figure 3—figure supplement 2*). While we cannot rule out the possibility that the inherent sampling bias of the extracellular recordings has led us to miss some less active M2 or SMC neurons that also display a substantial degree of activity reorganization, these observations suggested that large-scale functional network re-arrangements between the inferred global behavioral contexts are particularly prominent in the ACC.

## Within each global functional re-configuration of ACC neural ensemble, the representation of a behavioral strategy is further shaped by its local prevalence

We next investigated whether ACC neural dynamics are further shaped by more local adjustments to the animal's course of action. Indeed, within the otherwise largely stable functional ensemble configurations reflected in darker-colored squares of the similarity matrices (*Figure 3d*, 'within-dominant' and 'within-exploratory' squares), some local variation in pixel intensity was consistently observed. Therefore, we sought to determine whether the underlying variability in the ACC neural ensemble dynamics characterizing individual instances of sequence execution may reflect local fluctuations in the animal's choices. As a natural extension of defining *global* context (above) based on the roughly block-wise, persistent dominance of a single sequence, we chose the simplest summary statistic – sequence prevalence in recent choices – to capture *local* fluctuations in how much pursuit of the sequence in question is balanced by exploration of other strategies (Methods).

Firing rates indeed tracked local sequence prevalence in a substantial fraction of individual neurons that otherwise displayed a variety of response profiles (see *Figure 4a* for examples). Although a fraction of ACC neurons tracked local sequence prevalence throughout sequence execution, the majority did so in a subset of the five analysis windows that were aligned to nose port entries, roughly equally distributed across the temporal extent of the multi-step sequence (*Figure 4—figure supplement 1c*). To examine this modulatory influence on ACC activity in greater detail, we fit the relationship between the firing rates of individual neurons and the local sequence prevalence with a linear model – with or without the global context ('dominant'/'exploratory') as a fixed parameter – and evaluated the model's performance through cross-validation (Methods). The model's explanatory power was robust to the precise number of trials in the past used to calculate local sequence prevalence but was diminished if the estimate was shifted to largely comprise future trials (*Figure 4—figure supplement 1a–b*). The cross-validated performance of the mixed-effects linear model that included global context exceeded the performance of the model that had prevalence as a single parameter (*Figure 4—figure supplement 2a*), arguing that prevalence tracking and the large-scale functional reorganization of the ACC network between the 'dominant' and 'exploratory' contexts in our task reflect distinct computations. Strikingly, when we allowed the slope of the linear relationship between the strategy prevalence and the firing rate of individual neurons to vary depending on the global behavioral context, this simple behavioral summary statistic explained up to 80% of neural activity variance in individual ACC neurons during sequence execution (*Figure 4b and c*, *Figure 4—figure supplement 1d*; median 28, IQR 42% of activity variance explained by prevalence across all recorded cells in the rostral ACC). The marked modulation of individual cell activity in the ACC by local strategy prevalence stands in contrast with a more modest modulation detectable in other parts of the medial lobe, in particular, the SMC (*Figure 4b*, *Figure 4—figure supplement 2a*), pointing to a distinct role played by the rostral medial frontal cortical region in strategy contextualization. Within the ACC, an encoding potential gradient emerged following a detailed spatial reconstruction of the recorded units, with the most substantial modulation observed in the rostral-most portion thought to be homologous to the primate area 32D (*Figure 4b and c*). Thus, this underexplored, rostral-most portion of the ACC may play a particularly prominent role in keeping a statistical representation of recent behavioral choices.

The comparatively low amount of neural activity variance in the motor regions M2 and SMC explained by strategy prevalence suggests that the observed modulation is unlikely to reflect movement parameters that may co-vary with changes in the animal's dedication to a particular strategy. To establish this more explicitly, we directly quantified trial-by-trial measures of movement vigor (sequence execution time) and kinematics (the first principal component of movement trajectory) for each instance of sequence execution and then asked how well linear models that incorporate these parameters performed at explaining variance in ACC firing rates (Methods). We first evaluated

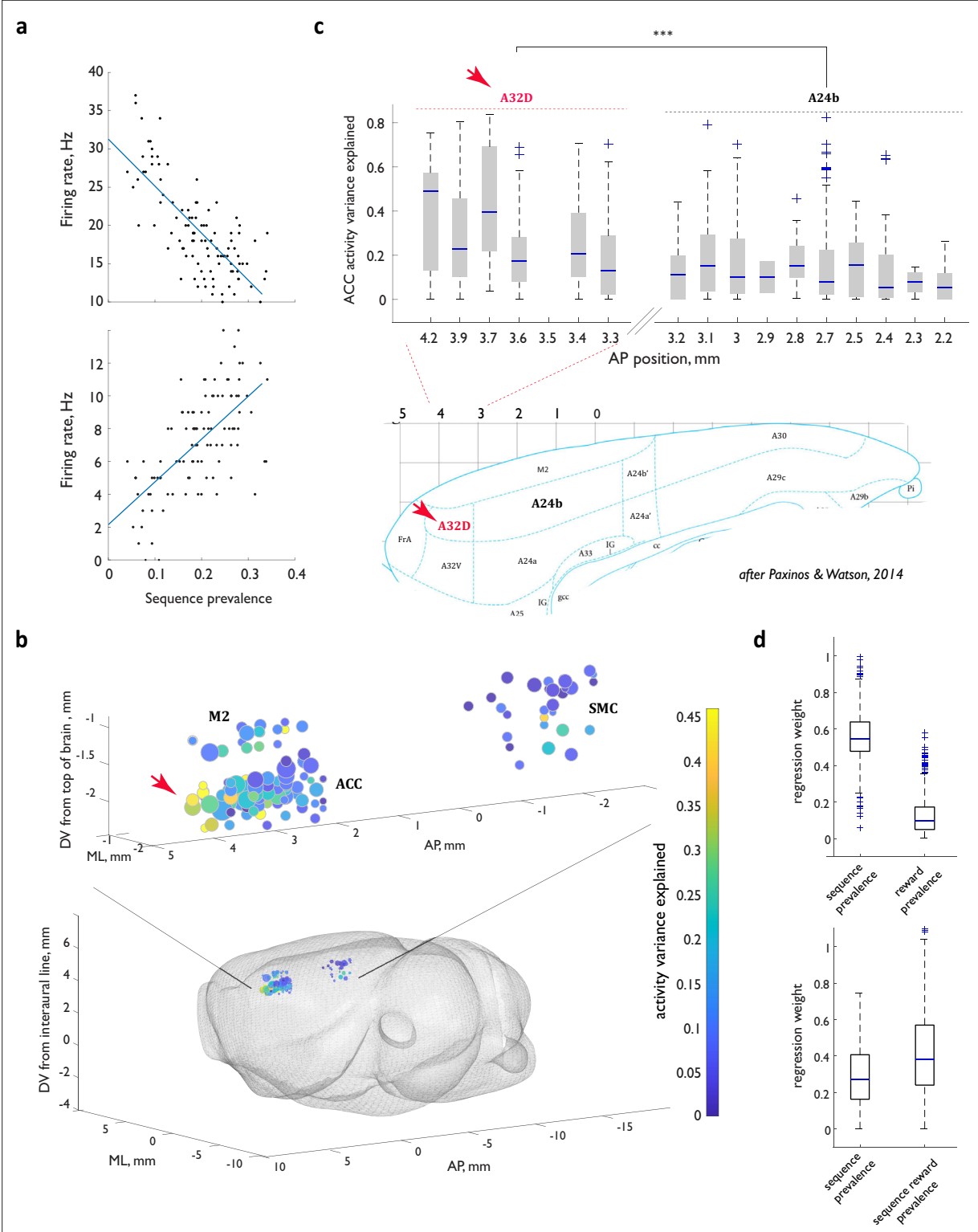

**Figure 4.** Activity of ACC neurons during sequence execution is markedly shaped by the local prevalence of the executed sequence. (**a**) Firing rates as a function of local sequence prevalence for two example ACC neurons. (**b**) Graphic representation of the explanatory power of the linear model across the spatial extent of the recording locations. (**c**) Top panel: explanatory power in the rostral portion of the cingulate as a function of location along the anterior- posterior axis. Bottom panel: refence rat brain atlas section. Red arrow in (**b, c**) points to a cluster of the particularly strong model performance in the rostral portion of the cingulate that maps to the region homologous to the primate area 32D. (**d**) Regression weights for the expanded linear models that relate ACC neuron FR rates to not only sequence prevalence but also general reward prevalence (upper panel) or sequence-specific reward

*Figure 4 continued on next page*

*Figure 4 continued*

prevalence (lower panel). For all box-and-whisker plots, central blue line indicates the median, the bottom and top edges of the box indicate the 25th and 75th percentiles, respectively, and the whiskers extend to the most extreme data points not considered outliers. n.s., not significant; **, p<0.01; ***, p<0.001.

The online version of this article includes the following figure supplement(s) for figure 4:

**Figure supplement 1.** Modulation of ACC activity is best explained by prevalence in recent (within tens of trials) past, robust across analysis windows and animals.

**Figure supplement 2.** Models relating neural activity to strategy prevalence retain robustness when trained on unrewarded sequence instances and display more robust performance in ACC than in other areas of the medial frontal lobe.

**Figure supplement 3.** Variance of movement vigor and trajectory, or local differences in other past choices cannot account for the robustness of sequence prevalence models in explaining ACC neural activity variance.

the performance of an equivalent linear model that used either execution time or the first principal component of movement trajectory as a single parameter; neither model could explain ACC neural activity variance as well as sequence prevalence (*Figure 4—figure supplement 3a*, 0.025+/–0.021 activity variance explained by movement trajectory, 0.04+/-0.04 by execution time, n=36 sessions, N=4 animals). We then focused in more detail on the dataset recorded in the anterior part of the cingulate cortex that displayed the strongest modulation by strategy prevalence (*Figure 4b and c*), and asked whether adding a separate movement-related parameter to the linear model would dwarf the explanatory power of the prevalence term (Methods). To ensure that the resulting weights on the two parameters would directly reflect their relative contributions, we z-score normalized all the variables to 0 mean and standard deviation of 1 prior to fitting the model. Consistent with the relatively poor performance of the single movement-related parameter models, the expanded models revealed only a minor contribution from those motor parameters to the overall model's performance (*Figure 4—figure supplement 3b*, absolute regression weights of 0.29+/-0.10 vs 0.11+/-0.07 for sequence prevalence and execution time; 0.32+/-0.12 and 0.08+/-0.04 for sequence prevalence and movement trajectory; n=36 sessions, N=4 animals). While it remains possible that more nuanced aspects of movement, like the animal's posture, contribute to the observed activity modulation, these observations suggest that gross movement-related parameters are not behind the explanatory power of strategy prevalence.

We also resolved whether the observed activity variation could simply reflect the specifics of choice and reward history that co-vary with prevalence. To accomplish this, we evaluated whether the prevalence-based model trained only on a subset of sequence instances that shared immediate history still had significant explanatory power. Specifically, we capitalized on the previous finding that much of such direct history impact in the ACC is exerted within fewer than five trials (*Bernacchia et al., 2011*). Since the prevalence estimate is formed over a much longer window, we could sub-select those 'LLR' sequence instances that were matched in the immediate history (i.e. 'LLR' instances that immediately followed another 'LLR'), but that otherwise still differed in the associated 'LLR' sequence prevalence. The prevalence-based model trained on this reduced subset of sequence instances matched for local history still explained a significant fraction of ACC activity variance (*Figure 4—figure supplement 3c*), arguing that the observed modulation could not have been solely history-based.

Overall, while our analyses do not rule out potential contributions of other trial-by trial behavioral parameters to neural encoding in ACC, they demonstrate that the gross motor and history parameters cannot account for the robustness of local strategy prevalence in explaining ACC neural activity variance.

## The tracking of reward by the ACC neural ensemble is strategy-specific, however reward prevalence is insufficient to account for activity modulation

Models of decision-making rarely include a summary statistic of the agent's recent behavioral choices – such as the local strategy prevalence that we examine here. This prompted us to next consider how the observed modulation of ACC ensemble activity by local strategy prevalence might interact with any modulation exerted by the external reward – a more commonly considered parameter directly related to the concept of valuation thought to engage circuit computations in the ACC (*Boorman*

*et al., 2009*; *Kennerley et al., 2009*; *Kolling et al., 2012*; *Luk and Wallis, 2013*). Indeed, variation in strategy prevalence is necessarily accompanied by variation in detailed reward history and thus understanding the interplay between the two in shaping ACC neural dynamics may shed further light on how the animals parse their environment.

We first established that local strategy prevalence provides at least some modulatory influence independent of external reward by evaluating the performance of the prevalence model trained exclusively on sequence instances executed under 'no reward' conditions. Specifically, we evaluated two distinct cases: 'exploratory' sequence instances in sessions comprising blocks of the familiar 'LLR' and 'RRL' targets, and persistent 'LLR' and 'RRL' sequence instances within the unfamiliar 'LLLR' and 'RRRL' contexts. The observed model performance in both cases was significantly more robust than what would be expected if most of the observed activity variance during sequence execution arose from the statistics of the associated reward (*Figure 4—figure supplement 2a*, 0.13 +/- 0.06 of ACC activity variance explained for exploratory sequence instances, n=36 sessions, N=4 animals; *Figure 4—figure supplement 2c*, 0.35+/–0.23, 0.25+/-0.19 and 0.24+/-0.19 A32D activity variance explained for unrewarded 'LLR' instances dominating early 'LLLR' acquisition in 3 individual animals). In further support of the conclusion that sequence prevalence shapes ACC neural dynamics independent of the associated reward, we observed little activity modulation for sequence instances within 'dominant' tails – a period when sequence prevalence remains high, but reward expectation should rapidly diminish (*Figure 3f*). Combined, these data argue that a summary statistic of that strategy in past choices exerts a continuous modulation of the strategy representation in ACC and raise the question of how this modulation interacts with any shaping of the ACC activity by the external reward.

ACC is known to both multiplex information about reward and individual actions (*Hayden and Platt, 2010*), as well as track progression to reward over multiple steps of an action sequence (*Shidara and Richmond, 2002*). To determine whether the unexpected robustness of the simple strategy prevalence model above in explaining a marked fraction of ACC neural activity variance during the execution of a specific sequence derives mostly from the impact of successfully procured reward, we incorporated an additional 'reward prevalence' term into the linear model and determined the relative contribution of the 'sequence prevalence' and 'reward prevalence' terms to the expanded model's performance (see below). Provided that the two parameters are sufficiently decorrelated in our dataset to avoid collinearity – due to factors like reward omission, exploratory sequence instances, and transient persistence with off-target sequences– their weights in the expanded model would reflect the relative contribution to the model's performance from the summary behavioral statistic itself and from the recently procured reward. To account for the possibility that reward statistics may be processed by, and exert influence on, the ACC circuitry at either the single trial- or the single sequence level, we separately considered a model with an 'overall reward prevalence' term and one with a 'sequence-specific reward prevalence' term. As such, a reward obtained for the target 'RRL' sequence before an exploratory, off-target, 'LLR' sequence instance would contribute to the 'overall reward prevalence' term, but not to the 'sequence-specific reward prevalence' term in models aimed at explaining neural activity variance during 'LLR' execution. We computed each 'reward prevalence' term by weighing the relevant previous rewards with the same temporal filter used to calculate sequence prevalence and identified a subset of behavioral sessions for which the 'sequence prevalence' and each of the 'reward prevalence' parameters were sufficiently decorrelated in our dataset (condition index between 1 and 4, Methods).

Does the observed marked modulation of ACC neural dynamics during different instances of sequence execution outside of some particularly persistent 'no reward' contexts arise largely from variation in recent reward history? To resolve this, we determined whether the model's weight for the 'sequence prevalence' term became negligible once the 'reward prevalence' term was added to the model. Contrary to this expectation, neither the 'overall reward prevalence' nor the 'sequence-specific reward prevalence' terms dwarfed the contribution from the 'strategy prevalence' term to the model's performance, with the 'overall reward prevalence' making a particularly small contribution to the model's explanatory power (*Figure 4d*). Furthermore, while the contribution from the 'sequence-specific reward prevalence' was on par with that of 'strategy prevalence', it was clearly insufficient on its own to account for the model's explanatory power (*Figure 4d*, regression weights of 0.558+/-0.005 vs 0.122+/-0.003 for sequence prevalence and overall reward prevalence; 0.285+/-0.008 and 0.416+/-0.012 for sequence prevalence and sequence-specific reward prevalence; n=47

sessions, N=9 animals). Combined, these observations argue that the tracking of reward by ACC is strategy-specific, and that both a summary statistic of the specific self-guided behavioral strategy in recent past and a statistic of the associated reward shape the ACC neural dynamics during strategy execution, further highlighting the central role of the animal's strategy choice rather than the external rule in this process.

## Strategy prevalence can be decoded from ACC ensemble trajectories during strategy execution

The unexpectedly strong modulation of ACC neural activity by sequence prevalence (*Figure 4*) points to the possibility that information about how prevalent the currently sampled strategy has been in the recent past may be decodable from ACC ensemble activity by downstream circuits. Indeed, we found that linear population decoders trained independently in each analysis window achieved robust cross-validated performance. For each of the five analysis windows, we fit the relationship between the local sequence prevalence and the firing rates – in that window – of all simultaneously recorded ACC neurons. Cross-validated performance of such individually tuned linear decoders scaled steeply with the number of neurons (*Figure 5—figure supplement 1a*). Moreover, ensembles of ACC neurons containing as few as 10 prevalence-modulated units could explain as much as 60% of sequence prevalence variance. Consistent with this finding, visualizing ensemble activity in a reduced dimensional state space defined from regression against prevalence (see Methods) revealed a clear separation of trajectories associated with different instances of sequence execution by the local strategy prevalence (*Figure 5a*).

The decoding strategy above implicitly assumes an ability to use a temporally varying readout, which may be challenging to implement in circuit dynamics. We therefore next examined the ability of temporally fixed readouts to decode prevalence. Focusing on sessions with at least five simultaneously recorded neurons that displayed modulation by prevalence in at least one analysis window, we next fit a new linear model that was constrained to always use the same readout (i.e. same weight for any given neuron in all five windows). The constrained model retained a large fraction of the overall explanatory power compared to that of the best of five individually tuned decoders (*Figure 5b*, left panel, variance explained of 0.36+/-0.03 for best of individually tuned models vs 0.23+/-0.02 for the model with weights fixed across the five analysis windows, n=36 sessions, N=4 animals). Furthermore, when we evaluated the prediction of this aggregate-constrained model in each of the five analysis windows, we observed substantial explanatory power even for the worst-performing window (*Figure 5c*, left panel, n=36 sessions, N=4 animals), suggesting that the fixed decoder would never drop the prevalence signal as the animal executed the sub-steps of a multi-step strategy. The ability of a fixed linear sum of activity across the recorded population to retain a substantial fraction of explanatory power throughout the temporal extent of sequence execution was not simply due to the presence of a small number of neurons with prevalence-related modulation in all five analysis windows. Indeed, the constrained model's performance dropped only moderately when such neurons were left out of the dataset (*Figure 5b–c*, right panels). Combined, these observations suggest that despite the transient nature of activity modulation by sequence prevalence at the level of individual neurons (*Figure 5—figure supplement 1b*), the ACC neural ensemble dynamics is structured in such a way as to permit this, or a related, summary statistic of the animal's recent sequence choices to be stably decoded by downstream circuits throughout sequence execution.

## Discussion

The Anterior Cingulate Cortex is thought to play a central role in dynamic, context-specific strategy arbitration in complex non-stationary environments, yet the organizing principles of the cingulate's neural activity that underpin contextually appropriate strategy selection and point to specific computations that take place in the ACC remain unresolved. We report here that when animals search a structured task space through self-guided exploration, a substantial fraction of ACC neural activity variance can be explained by the prevalence of individual behavioral strategies. This prevalence encoding – particularly enriched in the most rostral portion of the ACC homologous to the primate area 32D – is preserved through large-scale functional rearrangements of the ACC network between distinct global behavioral contexts and is evident even in the absence of any pairing between strategy execution

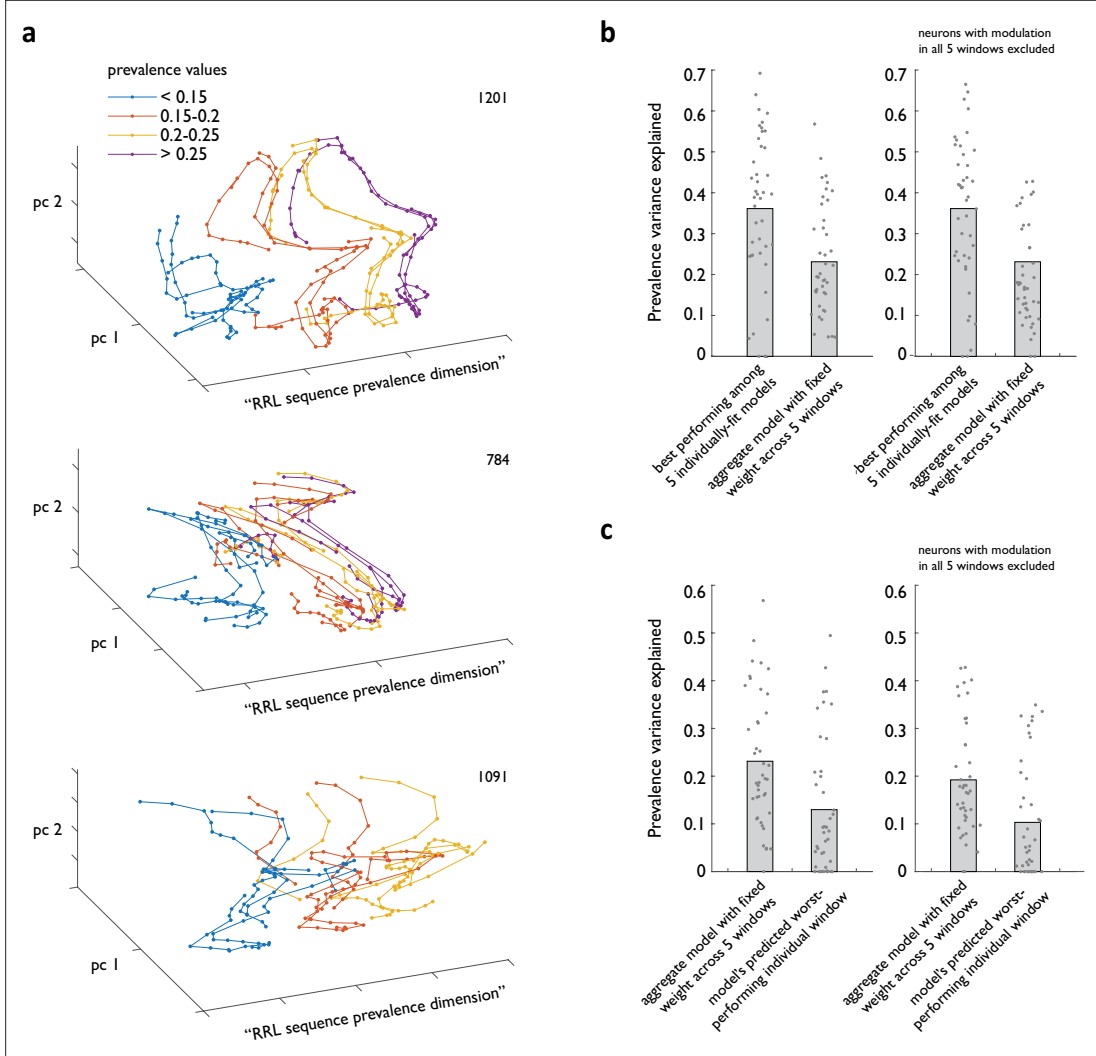

**Figure 5.** Strategy prevalence can be decoded from ACC ensemble activity throughout sequence execution. (**a**) Examples from three different animals of ACC ensemble trajectories associated with different instances of RRL sequence execution visualized in ensemble state subspace chosen to maximize separation by local sequence prevalence. (**b**) Cross-validated performance of linear models relating ACC ensemble activity during RRL execution to local RRL prevalence. Shown are the performance of best of five models fit in individual analysis windows (**b**), the average performance of model that was constrained to always use the same weight for any given neuron in all five windows (**b, c**), and that model's worst predicted performance in an individual analysis window (**c**). Right panels exclude from all models ACC neurons that show significant modulation by strategy prevalence in all five analysis windows.

The online version of this article includes the following figure supplement(s) for figure 5:

**Figure supplement 1.** Prevalence decodability scales with the ACC ensemble size, but rarely persists throughout the temporal extent of strategy execution at the single neuron level.

and reward delivery. We further show that strategy prevalence, or a related summary statistic of the specific self-guided behavioral strategy in recent past synergizes with a statistic of the associated reward to shape the ACC neural dynamics during strategy execution. Our findings raise the possibility that 'attention to *self*-action' (***Blakemore et al., 2000***) may be at the core of rostral cingulate functionality, in essence uniting the long-standing 'attention to action' account (***Norman and Shallice, 1986***; ***Passingham, 1996***) with the proposed role of the cingulate in processing information relating to self (***Blakemore et al., 2000***; ***Euston et al., 2012***; ***Passingham et al., 2010***; ***Seamans and Floresco, 2022***) and the production of self-generated actions.

Studies to define the precise role ACC plays in value-based decision-making have emphasized its role in tracking task-relevant information to guide appropriate action, but have otherwise often

cast the question in terms of identifying the specific step to which it contributes in the pre-decision computation and comparison of action values (*Boorman et al., 2009*; *Kennerley et al., 2009*; *Kolling et al., 2012*; *Luk and Wallis, 2013*). The resulting lack of a unifying account has prompted a recent suggestion that rather than contributing to pre-decision valuation, ACC encodes post-decision variables related to the subjective value that can be inferred empirically from the animal's choices (*Blanchard and Hayden, 2014*; *Cai and Padoa-Schioppa, 2012*). The notion of a subjective approach by each animal to the encountered task is similarly inherent in our parsing of the behavioral session using the animal's selected self-guided strategy (rather than the imposed rule) in examining the encoding of behaviorally relevant information. Our observation that strategy prevalence explains a substantial fraction of variance in the activity of ACC neurons supports the interpretation that ACC computes an inherently subjective post-decision variable, and indeed, subjective value and strategy prevalence are closely related in well-trained animals in typical value-based tasks. However, our finding that prevalence contributes strongly to activity modulation even in the absence of reward delivery suggests that prevalence encoding might be a more parsimonious account that accommodates both past observations and our findings. Examining how adding such a summary statistic of the agent's recent behavioral choices to models of decision-making changes their explanatory power in various experimental settings will be one interesting direction for future study.

The encoding of strategy prevalence appears to be independent of, and be preserved through, large-scale functional rearrangement of ACC networks. These functional rearrangements result in representational remapping, whereby ACC ensemble activity associated with the execution of a specific strategy changes markedly between behavioral contexts. This observation suggests that 'network resets' previously observed to accompany transitions to exploration (*Durstewitz et al., 2010*; *Emberly and Seamans Jeremy, 2019*; *Karlsson et al., 2012*) are not mere bookmarks of a behavioral state change, but rather reflect reorganizations of the ACC network designed to tag action plans with a global context-specific representation. Although 'dominant'/ 'exploratory' distinction maps well onto the 'default'/'alternative' dichotomy that has enjoyed prominence in the ACC field due to its computational appeal and explanatory power in many settings (*Blanchard and Hayden, 2014*; *Boorman et al., 2013*; *Kolling et al., 2012*; *Procyk et al., 2000*), the representational transitions in ACC likely reflect broader contextual tagging of action plans, as suggested by the observed difference in the representations associated with a specific exploratory sequence depending on the nature of the default strategy. It is even possible that the contextual content of the neural representations in ACC may not be limited to the statistics of actions and outcomes. Indeed, given the ACC's topological centrality within the frontal cortical network and domain-general role in cognition, the content of its neural representations set-up by large-scale re-arrangements of functional networks might come to reflect, when appropriate, distinct cognitive loads (*Shenhav et al., 2016*), social settings (*Tomlin et al., 2006*) or somato-visceral states (*Caracheo et al., 2018*; *Seamans and Floresco, 2022*). As such, ACC would set up – and possibly initially infer — representations of distinct task-relevant contextual information in such a way that tracking of individual strategy prevalence may then take place.

How is the circuit implementation of the abrupt representational re-organizations that define individual behavioral contexts related to prevalence encoding? One possibility is that these representational re-organizations are simply an emergent property produced by the dynamic interactions among the elements of the network that tracks prevalence. Specifically, a gradual, prevalence-tracking change in the activity of individual neurons could eventually reach a threshold that instantiates a sudden phase transition reorganizing ACC into a new functional network. Against this idea, prevalence often changes abruptly rather than slowly at the end of a behavioral block, and furthermore, changes in prevalence inside 'dominant' blocks often exceed those at the end of a behavioral block but do not lead to abrupt reorganizations of the network. Alternatively, the ACC network may be built to allow a continuous representation of prevalence through a substantial degree of reorganization and a different computation triggers the abrupt transitions. What constraints such an encoding scheme places on the ACC network architecture, where the transition-triggering computation is performed, and what factors into that computation remain open questions. Furthermore, investigating what constraints an encoding scheme that preserves prevalence computation through large-scale functional rearrangements places on network architecture and biophysics may likewise be an intriguing area of future study.

What might be the computational advantage of encoding prevalence, and how can the encoding of this post-decision variable be reconciled with the recent evidence causally linking ACC to the current decision (*Tervo et al., 2021*)? Given ACC's well-documented role in exploration (*Blanchard and Hayden, 2014*; *Fouragnan et al., 2019*; *Hayden et al., 2011*; *Kolling et al., 2012*; *Quilodran et al., 2008*; *Tervo et al., 2021*), one possibility is that keeping track of strategy prevalence may be used to prioritize exploring strategies evaluated less frequently in the recent past (see *Wiering and Schmidhuber, 1998* for one implementation of such frequency-based exploration). An alternative view is informed by old accounts of the ACC's role in recognizing the 'self' actions (*Blakemore et al., 1998*; *Espinosa et al., 2006*). Indeed, an intriguing possibility is that in complex settings, keeping track of different actions taken and the frequency with which one has performed them might be an effective way of estimating agency – an understanding of which actions bring about the ability to promote or prevent the occurrence of events in the environment – through statistical learning. As such, agents can determine the degree to which their actions exert control over world events without reinforcement or an explicit representation of the temporal relationships between their actions and events. Statistical learning is thought to be central to many aspects of perceptual cognition and motor control; in the causal domain, it provides a complement to associative learning in permitting the agent to make inferences about the world without being enslaved to temporal contiguity or specific action-outcome contingencies. In principle, learning of statistical regularities can happen implicitly – without explicit awareness or hypothesis testing. However, one influential view posits that behind the efficiency, with which animals pick up on statistical regularities from sparse data in complex, often only partially observable environments, is the process of probabilistic inference that entertains multiple candidate models of the underlying statistical regularities coupled with explicit hypothesis testing designed to evaluate those models (*Tenenbaum et al., 2011*). And indeed, establishing agency would critically depend on active evaluation of an inferred ability to influence events in the world. We posit that ACC's central role in establishing one's agency is what reconciles our finding of its robust prevalence encoding and the causal evidence (*Tervo et al., 2020*) linking it to the ongoing decision of whether or not to evaluate alternative action plans (and more generally to curiosity *Wang and Hayden, 2020*, information seeking *White et al., 2019* and hypothesis testing *Elliott and Dolan, 1998*).

We note that the interpretation that ACC – and likely the medial frontal network into which it is embedded – plays a key role in evaluating the extent to which one has agency in the environment is a refinement of the broader hypothesis that it processes information related to self and implements self-generated actions that not only accommodates the existing experimental observations but is also more falsifiable. Indeed, a major critique of the 'self-generated actions' view of the medial frontal lobe is that the distinction between self-generated and externally triggered actions is empirically intractable, thus making this view of the medial frontal lobe unfalsifiable (*Schüür and Haggard, 2011*). In contrast, it is conceivable to develop experimental designs that manipulate the reward statistics to induce miscalculations of agency: superstitions – a perception of control over the environment in the absence of a causal link between the agents' actions and the outcome, and learned helplessness – an incorrect perception of a lack of control and a cessation of action.

## Methods

### Subjects

All experiments were done in male Long Evans rats 6–12 months of age (with weight kept between 400 and 500 g). Animals were kept at 85% of their initial body weight before food restriction and maintained on a 12 hr light/12 hr dark schedule. Experiments were conducted according to National Institutes of Health guidelines for animal research and were approved by the Institutional Animal Care and Use Committee at HHMI's Janelia Farm Research Campus (IACUC protocol 22–0220.02).

A total of 7 animals were implanted with tetrode drives for collection of neural activity during the task (4 animals targeting the ACC and 3 animals targeting the SMC).

### Behavioral apparatus and task

All behavior was confined to a box with 23 cm high plastic walls and stainless-steel floors (Island Motion Corp). The floor of the box was 25 cm by 34 cm, and the nose ports were all arranged on one of the 25 cm walls. All lights, nose ports, and reward deliveries were controlled and monitored with a

custom-programmed microcontroller, which in turn communicated via USB to a PC running a Matlab-based control program. Nose port entries were detected with an infrared beam-break detector (IR LED and photodiode pair). The central initiation port contained one white LED that indicated the option to initiate a new trial. The side choice nose ports also each contained an LED that indicated that the initiation port had been successfully triggered (at which point the LED in the center port was extinguished) and side ports were available for selection. Note that in some sessions, only the center LED was changing states, and in some, no lights were used with little impact on behavioral performance. The side ports also delivered liquid rewards (0.1 ml drops of 10% sucrose mixed with black cherry Kool-Aid) with the help of a motorized syringe pump (Harvard Apparatus PHD 2000).

The behavioral task is an elaboration of the basic design reported as a 'covert pattern' task in *Tervo et al., 2014* and consists of a series of self-initiated trials (several hundred per session), each involving a choice between two options – the left and the right choice port. Reward is delivered on the last step of a target sequence that is not otherwise indicated to an animal and thus has to be discovered through self-guided exploration.

Each session in the dataset described in this manuscript contained one to several unsignalled transitions in the identity of the target sequence. The set of possible sequences typically included some or all of the four non-trivial three-step sequences ('Left-Left-Right', 'Right-Right-Left', 'Left-Right-Right', 'Right-Left-Left'), but occasionally was expanded to include longer sequences (mainly 'Left-Left-Left-Right'). Only one sequence was rewarded in any particular block of trials. A subset of sessions incorporated reward omission for 10–30% of correctly executed sequences.

## Behavioral training

Food-restricted animals were trained to perform the task with no explicit instruction. Early in training, exploration was encouraged by rewarding a small fraction (at most 10%) of novel patterns – specifically, those that escaped prediction by Competitor 2 (*Tervo et al., 2014*) that used the history of the animal's choices and outcomes to predict his next choice. Note that once the animal discovers and concatenates the target sequence at a high rate, little to no background reward is delivered since the behavior becomes fully predictable. Eventually, background reward was fully eliminated.

Animals were considered proficient on the task when they consistently discovered and concatenated more than one sequence in a session, and when the average across-session reward rate was consistently in the 17–22% range % (with 33% being the maximal theoretically possible). Over 90% of animals became proficient within one month of training.

## Electrophysiological recordings

A microdrive array containing 16 independently movable tetrodes was chronically implanted on the head of the animal. Each tetrode was constructed by twisting and fusing together four insulated 13 µm wires (stablohm 800 A, California Fine Wire). Each tetrode tip was gold-plated to reduce impedance to 200–300 kΩ at 1 kHz. Within the implant, the tetrodes converged to an oval bundle (1 mm x 2 mmd), angled at 0° with respect to vertical (pointing towards midline after implantation).

For the drive implantation surgery, trained animals were initially anesthetized with 5% isoflurane gas (1.0 L/min). After 10–15 minutes, isoflurane was reduced to 1.5–2.0% and the flow rate to 0.7 L/min. A local anesthetic (Bupivacaine) was injected under the skin 10 minutes before making an incision. A unilateral craniotomy (1.0 by 2.0 mm) was drilled in the skull above the site of recording. The microdrive array was implanted such that the tetrode bundle was centered 3.0 mm anterior and 0.8 mm lateral to Bregma (right or left hemisphere) when recordings were targeted to the ACC, and 2.0 mm posterior and 1.2 mm lateral to Bregma when recordings were targeted to the putative SMC. Small stainless steel bone screws and dental cement were used to secure the implant to the skull. One of the screws was connected to a wire leading to the system ground. Before the animal woke up, all tetrodes were advanced into the brain ~1.20 mm deep from the brain surface.

Over the two weeks following surgery, the tetrodes were slowly lowered, moving approximately 40 µm/day on average. During this time, animals were re-acclimated to performing the task with the drive. When performance on the task was regained to pre-surgery levels (in terms of motivation and dynamic strategy arbitration behavior), recording sessions began. After each recording session, any tetrodes that did not appear to have any isolatable units were moved down 25 µm. Once a tetrode

had been moved a total of 2.5 mm from the surface, which is the approximate border between anterior cingulate and prelimbic cortices, it was no longer advanced.

Each recording session spanned 3–4 hr. Animals were not forced to perform the task and sometimes took breaks (generally around 10 min, but sometimes up to 30 min). Data from all the animals were collected using the wireless headstage and datalogger (Horizontal Headstage 128ch with Datalogger, SpikeGadgets, spikegadgets.com/hardware/hh128.html). An array of LEDs of different colors was attached on top of the animal's implant and the animal's position in the environment was recorded with a video camera at 60 frames per second. The animal's position was reconstructed offline using a semi-automated analysis of digital video of the experiment with custom-written software.

Raw electrophysiology data were sampled at 30 kHz, digitally filtered between 600 Hz and 6 KHz (2 pole Bessel for high and low pass) and threshold-crossing events were selected for further analysis. Individual units on each tetrode were identified by manually classifying spikes using polygons in two-dimensional views of waveform parameters using custom-made Matlab scripts (*Karlsson et al., 2012*, MatClust, https://www.mathworks.com/matlabcentral/fileexchange/39663-matclust). For each channel of a tetrode, peak waveform amplitude and the waveform's projection onto the first two principal components were used for clustering. Autocorrelation analyses were done to exclude units with non-physiological single-unit spike trains. Only units where the entire cluster was visible throughout the recording session were included. Thus, a unit was not isolated for further analysis if any part of the cluster vanished into the noise or was cut off by the recording threshold.

The total contribution from each animal was as follows:

ACC animals: 4 sessions, 81 neurons for animal 1, 15 sessions, 163 neurons for animal 2, 13 sessions, 324 neurons for animal 3, 5 sessions, 286 neurons for animal 4
SMA animals: 11 sessions, 176 neurons for animal 1, 12 sessions, 101 neurons for animal 2, 8 sessions, 75 neurons for animal 3

## Mapping of the putative SMC homologue

The Supplementary Motor Cortex had not been characterized in the rat, so in a separate set of experiments, we sought to identify a medial premotor region involved in temporally sequencing self-initiated actions in the rat. Specifically, we performed a set of pharmacological inactivation experiments along the rostro-caudal axis of the agranular premotor region M2, evaluating the effect of such a transient inactivation on the animals' performance during the three-step version of the sequence exploration task. We observed a robust behavioral effect following a bilateral injection of muscimol, but only when muscimol was delivered to the region immediately caudal to mid-cingulate cortex (a location similar to that of primate SMC, *Figure 3—figure supplement 2*). Following local muscimol delivery, animals continued to complete trials, but no longer performed the target sequence significantly above the value expected for a biased coin. SMC inactivation appeared to specifically affect complex sequencing rather than chaining of actions in general, because the animals still performed the sequential entries into the initiation and reward ports correctly. The technical details associated with this set of experiments are provided below.

## Surgery for Cannula Implantation

Location of bilateral craniotomies and cannula implantation were +2.0 mm AP and ± 1.2 mm ML with respect to Bregma for ACC, –2 mm AP and ± 1.2 mm ML for putative SMC,+1 mm AP and ± 1.2 mm ML for FOF (*Erlich et al., 2011*), –0.5 mm AP and ± 1.2 mm ML for another candidate SMC region that didn't show muscimol effect. ACC cannula implantation was deeper (2.0–3.)0 mm to account for curvature, whereas candidate SMC regions were 1 mm deep. Injection guide cannulas (Eicom CXG (T) 2 Diameter OD/ID 0.3/0.2 mm) were inserted through a 0.5 mm craniotomy. As a protection measure for the animal in the home cage, an opaque cone was placed around the implant. Cannulas were bonded to the skull with dental cement (C&B Metabond - Parkell). Dummy cannulas (Eicom CXD (T) 2 Diameter 0.15/0.06 mm) were used in between sessions to protect the cannula from debris. Food deprivation and further training were resumed after the recovery period.

## Muscimol inactivation

As a control, the beginning of each session remained unperturbed and animals were allowed to perform the task normally. After an animal performed 200 trials and reached reward rate above 0.22, the animal was placed back into its home cage for inactivation. Treats were provided to keep the animal still. Muscimol or saline was administered through the needle inserted into the cannula. Muscimol (Tocris Bioscience) solution was prepared with sterile saline (Hospira) at a final concentration of 0.1 μg/ml. Bilateral infusion was made via a 3 mm-long 31-gauge injector connected to a 5 μl syringe (Hamilton) by a teflon tube (Eicom). Eicom micro syringe pump ESP-64 was programmed to deliver the solution at the rate of 0.25 μl/min for 2 min, for a total volume of 0.50 μl for each hemisphere. The animal was placed back into the operant chamber after the injection. The muscimol effect persisted for about 2 hr after the injection. Trials within one hour of the injection were used for the analysis.

## Data analysis

### Session selection

Since animals frequently showed greater variability in behavioral performance following drive implantation, a minimal selection filter was applied to determine, which sessions to include into the analysis: the session had to contain at least 200 instances of either the 'LLR' or the 'RRL' sequence, and the across-session average sequence production rate had to exceed 0.15.

### Selection of dominant and exploratory sequence instances in behavioral data

The nature of our behavioral framework required extra care when parsing the behavioral record to identify 'legitimate' sequence instances. Parsing a continuous stream of left and right choices to identify 'legitimate' sequences was easy for the 'dominant' condition, where a high prevalence of sequence concatenation, and a scarcity of choices that conform to other patterns, argue that almost every instance of a pattern conforming to the target sequence is likely to be one actually evaluated by the animal. Specifically, starting with all 'LLR' ('RRL') instances that were done during the epochs where 'LLR' ('RRL') was the latent target sequence, we removed all examples prior to when the dominance for the new sequence (which we define as the presence of a successful concatenation of at least three sequence instances) hadn't yet been established, and appended all sequence instances from the period where dominance had persisted following a block change (i.e. until the previously concatenated sequence was interrupted by more than two trials incompatible with that sequence).

The 'exploratory' condition, required a closer examination outside of clear concatenations. For example, although some 'RRL' patterns within runs of 'LLRRL' in a 'LLR' block might reflect a true pairing of the locally dominant pursuit of 'LLR' with a quick exploratory evaluation of 'RRL' tagged on in a manner akin to strategy mixing (*Donoso et al., 2014*), most of such instances likely reflect a mere apposition of the 'RRL' and the 'LR' sequences. To select the likely true exploratory sequences conservatively, we therefore excluded all putative lone exploratory sequences that displayed an overlap (to the left or to the right) with the dominant sequence, *except* when the timing analysis (see below) provided independent evidence in favor of classifying this putative instance as a true exploratory sequence. Here, our approach was grounded in the expectation that animals would pause, if only briefly, at the side nose port on the last step of a true exploratory sequence instance to ascertain that the explored sequence is not rewarded. Under this assumption, we 'rescued' the putative exploratory sequence instances for which the duration of the third step exceeded the threshold chosen on the basis of an unbiased analysis of the distributions for the within-sideport and side-to-center durations across all 'dominant' sequence instances for which the otherwise scheduled reward was omitted permitted an unbiased selection of the specific threshold (*Figure 1—figure supplement 2e*). Specifically, starting with all putative 'LLR' ('RRL') during periods when dominance had been established for another sequence (see above), we first removed all instances that displayed an overlap (to the left, or to the right) with the dominant latent target sequence. From the removed set, we then 'rescued' those instances that were either preceded by another 'LLR' ('RRL'), or for which on step 3, the within-sideport duration exceeded 0.5 s, or side-to-center duration exceeded 1 s.

In addition, we noticed that at times, animals chose to concatenate 'LLRR', possibly as a clever way to catch every three-step sequence that might serve as the latent target. Therefore, we removed

from all datasets any 'LLR' ('RRL') instances that appeared within such concatenations. Finally, we also removed from all datasets sequence instances where the animal appeared to take long breaks. Specifically, any sequence instance that displayed a center-to-side time of over three seconds was removed from further analysis.

## Analysis of sequence representation in frontal cortical activity

To capture the neural activity associated with a specific three-step sequence, we represented, for each neuron, its activity during a single instance of sequence execution as a vector of the square root of spike counts in five 500 ms windows centered on port entry events (center and side port entries for steps 2 and 3 in the sequence, and side entry for step 1; the window leading to center port entry in step 1 was excluded to avoid a strong contribution of behavioral choices that preceded sequence initiation). The choice of working with the square root of spike counts was guided by the desire to transform the data with Poisson distribution into an approximately Gaussian distribution with a unity variance (*Anscombe, 1948*). Such transformation brought the variance of the binned data for individual cells to the same level, such that the variability of fast-spiking neurons did not conceal less active cells. The five chosen windows offered substantial coverage of the sequence with minimal temporal overlap, a high degree of stereotypy of spatial trajectories, and the exclusion of outcome-related activity.

For a single neuron, its activity associated with each instance of sequence execution was thus captured as a single point in the corresponding five-dimensional 'state space'. A single-cell metric of the contextual representational change ('transition score', *Figure 3*) was calculated as the Euclidean distance in this single cell state space between the centroids of the 'dominant' and the 'exploratory' clusters.

## Decoding analysis

For each session, we trained a linear classifier to predict, on the basis of the ensemble activity, whether a particular trial was a part of the 'LLR' or the 'RRL' sequence. For each trial, we used firing rates of simultaneously recorded ACC cells as predictors and sequence identity as a predicted variable. All cells with non-zero firing rate were included in the analysis; the firing rate for each was calculated over the 500 ms decoding window centered around entry into the side port. When the entire three-step sequence identity was being decoded, each cell contributed three windows corresponding to the three side entries. When the decoding was done on the context ('LLR' vs 'RRL') of a specific choice (e.g. 'Right'), only one window was used. All linear discriminant analyses were performed with a five-fold cross-validation: five separate classifiers were trained on 80% of the data set, and the classification accuracy for each was estimated on the remaining 20% of the dataset, after ensuring that the sample size in each category was the same. The reported values represent the average error rates across the five classification runs.

## Characterization of representational transitions

To capture the behavior of the entire neuronal ensemble, the dimensionality of the state space was expended to 5*N, where N was the number of simultaneously recorded cells. We then calculated pairwise Euclidian distance between the points in the state space to evaluate how the sequence representation evolved over the course of the session. To demonstrate clustering of points according to the behavioral context, we calculated the distance between centroids of the clusters (median values along each dimension) corresponding to the 'dominant' and 'exploratory' groupings, and compared it with the corresponding distance calculated after context identity was randomly shuffled across the differences sequence instances.

## Characterization of prevalence encoding

To generate a running estimate of the local sequence prevalence in the animal's behavioral choices, we constructed, for each behavioral session, a vector of binary values of the same length as the session. Trials corresponding to the last step of the target sequence were given the value of '1' (regardless of the context), all other trials were given the value of '0'. This binary vector was then convolved with a causal half-Gaussian kernel with zero mean and standard deviation of 20 trials.

We used fivefold cross-validated performance of a linear model to characterize the extent to which sequence prevalence factored into the activity of ACC neurons, using built-in Matlab functions 'fitlme' and 'crossval', and separately fitting the data in each behavioral context. Linear models were fit independently for each of the five analysis windows. For reporting fractions of cells with significant explained variance in each window, all five values for each neuron were used. For reporting explained variance of individual neurons, best of five values was used.

We defined explained variance as the coefficient of determination (R-squared) using the equation:

$$1 - \frac{\sum \left(y - y_{fit}\right)^2}{\sum \left(y - \bar{y}\right)^2}$$

(1)

where the top is the sum of squared errors and the bottom is the total variance. The explained variance was calculated for 'test' subsets of data for each of the five cross-validation rounds, and then averaged across the five resulting values. In some cases, the explained variance was negative reflecting overfitting (typically occurred in the data sets with a small sample size). These negative values were replaced with zeros for subsequent reporting.

## Interaction of prevalence and context

We considered several linear mixed-effect models to examine how a potential interaction of prevalence and context might account for the observed variance in sequence representation. The models considered incorporated sequence prevalence as a fixed effect predictor and could include a random intercept that varied by context. We used each of these features alone or together to fit the different models, and similarly estimated performance as the fraction of explained variance with fivefold stratified (with respect to context) cross-validation.

## Comparison with models relating movement vigor and trajectory to neural activity

We resolved whether the observed modulation of ACC neural activity could be explained by variation in sequence timing or spatial trajectory by evaluating how well linear models using these parameters as regressors performed relative to the models above. For a measure of movement vigor, we calculated the time between the first step 'center in' and the last step 'side in'; for the spatial trajectory, we first took snippets of the X and Y coordinates within the same five analysis windows that were used for all analyses of neural activity and then concatenated those snippets for all windows and both coordinates together in order to get a 1D vector. Then we combined such vectors for all the sequence instances within a session into a matrix and performed PCA to reduce the dimensionality of the raw spatial data but preserve the data about variation in animal motion. The first principal component was then used as a regressor for the corresponding linear model.

## Multiple linear regression

To determine whether the unexpected robustness of the simple strategy prevalence model in explaining a marked fraction of ACC neural activity variance during the execution of a specific sequence derives mostly from the impact of successfully procured reward or the associated movement parameters, we incorporated an additional term into the linear model and determined the relative contribution of the 'sequence prevalence' and 'reward prevalence' terms to the expanded model's performance. We considered several families of multiple linear regression models (such as 'reward prevalence', 'sequence reward prevalence', 'sequence execution time' and 'trajectory PC1'). We computed 'reward prevalence' and 'sequence reward prevalence' in a similar way to 'sequence prevalence' (see above) by convolving a binary vector of rewards (1's for rewarded and 0's for unrewarded trials) with half-Gaussian kernel with zero mean and standard deviation of 20 trials. The key difference between the two types of reward prevalence is that we assigned '1' to either all rewarded trials (for calculating 'reward prevalence') or only to trials rewarded for the specific sequence type (for calculating 'sequence reward prevalence'). We used the same variables for the movement vigor and spatial trajectory as described above. We whitened both the predictor and the response variables to zero mean and standard deviation of 1. We then fitted a multiple linear regression model using

built-in Matlab functions 'fitlm' with pairs of variables (i.e. sequence prevalence and another feature) as predictors and the firing rate of individual neurons in a certain window as a response variable. We used absolute values of the resulting predictor variable weights as a measure for the relative contribution to the final fit by each predictor (*Figure 4d* and *Figure 4—figure supplement 3b*). These fits were done for each of the five behavioral windows and the window with the highest total variance explained was selected for the assessment of weights. The following two criteria were used for the sub-selection of sessions and neurons in this analysis:

1. To ensure that the estimate of weights was not exquisitely sensitive to the specific sample, we only chose sessions whether the pair of variables in the model was sufficiently decorrelated. Specifically, we only took sessions that had condition index – a measure of decorrelation calculated as the square root of the ratio between the largest and the smallest eigenvalue of the covariance matrix – below 4.
2. For any chosen sessions, we sub-selected those neurons that were deemed significantly modulated by sequence prevalence in the initial analysis (variance explained >0.2). This sub selection was done to answer the question of whether any robustness in explaining activity variance came derives from another variable.

## Decoding sequence prevalence from ACC ensemble trajectories

To characterize the extent, to which sequence prevalence could be decoded from population activity buy a downstream decoder, we evaluated the 5-fold cross-validated performance of several linear models that used the firing rates (or more specifically, the number of spikes) of individual neurons in each of the five analysis windows as regressors to estimate the local prevalence of the sequence in question during the t-th instance of sequence execution. The most general model (*Equation 2*) modeled the response variable local sequence prevalence at the time when the t-th sequence iteration was executed as a linear sum of predictor variable set, with each neuron contributing 5 spike counts, each with its own regression coefficient (weight; 'model i, k').

$$Seq\ prev_t = bias + \sum_{i,k} w_{ik}\ spikes_{ikt} \tag{2}$$

where t=sequence instance, i=1:number of neurons, k=1:5 analysis windows.

For this most general model, each neuron received a significance score for each of the five analysis windows that showed whether that particular neuron in that particular window contributes to the prediction of sequence prevalence. We used these scores to sort each neuron into classes 0–5, depending on the total number of windows this neuron was significant in. Neurons that were not significant in any of the windows ended up in class 0 and were removed from all further modeling, and only sessions with at least 5 neurons outside of class 0 were retained for further analyses.

We further sometimes fit models on all neurons from classes 1 through 5, and sometimes fit models only on neurons from classes 1 through 4. The latter was done to guarantee that we had removed those neurons that themselves had significant prevalence encoding in all windows, thus forcing the models to use multiple neurons to output good predictions over the temporal extent of sequence execution.

To estimate the best possible prediction in each of the five analysis windows, we also fit five individual-window-models (model 'k'):

$$Model\ k: \quad Seq\ prev_t = bias + \sum_{i} w_{ik}\ spikes_{ikt} \tag{3}$$

Finally, we fit a linear model that was constrained to always use the same readout (i.e. same weight for any given neuron in all five windows, *Equation 3*, model $w_{ik} = w_{il}$). In essence, this single aggregate model treats each of the five windows as extra data points for the same fit. As such, while having the same number of model parameters, this model benefits from these extra data but loses the temporal resolution, providing an estimate of how the fixed downstream decoder would perform on average during any instance of sequence execution.

## Acknowledgements

We thank Elena Kuleshova for help with surgeries and histology. We thank Shaul Druckmann, Gowan Tervo, Michael Brainard and Vivek Jayaraman for advice and comments on the manuscript. Funding: This work was supported by the Howard Hughes Medical Institute.

## Additional information

### Funding

| Funder | Grant reference number | Author |
|---|---|---|
| Howard Hughes Medical Institute | | Alla Karpova |

The funders had no role in study design, data collection and interpretation, or the decision to submit the work for publication.

### Author contributions

Mikhail Proskurin, Conceptualization, Data curation, Software, Formal analysis, Validation, Investigation, Visualization, Methodology, Writing - original draft; Maxim Manakov, Conceptualization, Data curation, Software, Formal analysis, Validation, Investigation, Visualization, Methodology, Writing - original draft, Writing - review and editing; Alla Karpova, Conceptualization, Resources, Data curation, Supervision, Funding acquisition, Investigation, Methodology, Writing - original draft, Project administration, Writing - review and editing

### Author ORCIDs
Mikhail Proskurin (iD) http://orcid.org/0000-0003-2548-9722
Alla Karpova (iD) http://orcid.org/0000-0001-5869-6336

### Ethics

All animal experiments were conducted according to National Institutes of Health guidelines for animal research and were approved by the Institutional Animal Care and Use Committee at HHMI's Janelia Research Campus.

### Decision letter and Author response
Decision letter https://doi.org/10.7554/eLife.84897.sa1
Author response https://doi.org/10.7554/eLife.84897.sa2

## Additional files

### Supplementary files
• MDAR checklist

### Data availability

All data can be found here: https://doi.org/10.25378/janelia.21594129.v2. All code can be found here: https://doi.org/10.25378/janelia.21594105.v1. Requests for raw materials should be addressed to AYK (alla@janelia.hhmi.org).

The following datasets were generated:

| Author(s) | Year | Dataset title | Dataset URL | Database and Identifier |
|---|---|---|---|---|
| Proskurin M, Manakov M, Karpova A | 2023 | Dataset supporting main results of "ACC neural ensemble dynamics are structured by strategy prevalence" | https://doi.org/10.25378/janelia.21594129.v2 | Janelia Research Campus, 10.25378/janelia.21594129.v2 |

*Continued on next page*

*Continued*

| Author(s) | Year | Dataset title | Dataset URL | Database and Identifier |
|---|---|---|---|---|
| Proskurin M, Manakov M, Karpova A | 2022 | Analysis code supporting main results of "ACC neural ensemble dynamics are structured by strategy prevalence" | https://doi.org/10.25378/janelia.21594105.v1 | Janelia Research Campus, 10.25378/janelia.21594105.v1 |

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
