## [Editor Report]

This manuscript posits a novel role for the anterior cingulate cortex (ACC) in coding for sequential action strategies and the prevalence of each strategy. These findings provide important insight into ACC function and will therefore be of broad interest within the field of cognitive neuroscience. The evidence supporting the primary hypothesis is convincing.

---

## [Decision Letter]

**Decision letter after peer review:**

Thank you for submitting your article "ACC neural ensemble dynamics are structured by strategy prevalence" for consideration by *eLife*. Your article has been reviewed by 3 peer reviewers, and the evaluation has been overseen by a Reviewing Editor and Timothy Behrens as the Senior Editor. The reviewers have opted to remain anonymous.

Essential revisions:

The reviewers were mostly positive about the manuscript, but all agreed that the evidence ruling out alternative explanations should be strengthened. The high-level summary of the changes requested is as follows:

– The most significant weakness is insufficient evidence to rule out alternate hypotheses, particularly reward, but also the certainty and other factors that may account for the variance of neural signaling.

– More data on behavior is needed: accuracy on all strategies, how many blocks per session, did rats have individual biases in strategy, etc.

– Analytical methods are hard to follow in some places.

– Too much unnecessary jargon reduces clarity.

– Reinforcement effects not sufficiently integrated/described.

– Results and a more general description of the analytical methods should be in the main text rather than the methods.

– A more incremental and linear exposition would help readers understand the data and analyses.

*Reviewer #2 (Recommendations for the authors):*

I have some concerns and suggestions to make for this manuscript.

1) The authors claim the performance of the rats to be 'robust.' Is there an objective measurement to test such robustness? The conditioned probabilities in Figure 1b might provide some evidence. But it is not straightforward to comprehend what a probability of 0.4 means relating to behavioral performance. It would be helpful to have a learning curve showing baseline behavioral performance and whether expert rats have reached their behavioral asymptotes.

2) To identify exploratory sequence instances in the behavioral data, the authors try to remove many action sequence instances that could be explained by factors other than 'exploration.' While this effort is appreciated, it remains to be answered whether one could pinpoint the actual 'exploratory' instances or their existence. To do so, we need to figure out if a mistake made by rats is random, based on a false belief, or only caused by disengagement. Even if there are some exploratory sequence instances in the behavior, other causes might contaminate identified instances, which could promote the current findings on the coding of strategy prevalence. For instance, the authors use a criterion combining the within-side duration and side-to-center duration in step 3 to remove overlapped and long-break sequences. It, however, can only partially rule out other possible contamination as many trials display long choice duration even in step 1 (Figure S3), and the variability is also substantial. One suggestion is that the authors try a more stringent criterion and test whether the primary findings hold firmly.

3) The authors try to rule out many alternative explanations of the neural data other than strategy prevalence. But unfortunately, it seems impossible to remove all influences from some factors (such as action steps, action sequence configuration/composition, behavioral confidence, reward delivery, reward expectation, reward rate, expected reward location, etc.) that covary with strategy prevalence. (i) The authors examine neural responses to 'R' in LLR vs. RRL. But apparently, the two Rs are in different steps (step 3 in LLR and step 1 or 2 in RRL) within the two three-step sequences. Steps thus might explain such differential responses to the R in different sequences. Many neurons would likely show selectivity to different steps regardless of specific actions (L or R), which need further testing. (ii) The authors have tried to remove the influence of reward expectation by only looking at different exploratory states since no reward is available in these trials. But reward expectation can still explain differential firing to the dominant vs. exploratory strategies. The dominant tail effect seen in Figure 3f following the change of rules does not help because reward expectation, like strategy prevalence, also follows the subjective belief of the rats but not the actual task rule. (iii) ACC neurons might code an action sequence as a whole or configuration, much like a compound action, which might instead be interpreted as the authors' specific content of strategy prevalence. However, further evidence is needed to rule out the possibility that neural activities are merely related to a complex action per se rather than a higher cognitive cause. While it is understandable that confounds are not easy to control fully, the authors should emphasize these limitations more in the discussions.

4) Recording data show that neural correlates of strategy prevalence in M2 are slightly higher than in SMC, but inactivating SMC but not M2 affects corresponding behavior. How do the authors explain the disagreement between recording and inactivation data? More importantly, behavioral data on the inactivation of ACC is missing, although cannula implantation surgery in ACC is mentioned in the manuscript. More clarifications are needed for these questions.

5) In Figure 3e, data from a small subset of sessions show that neurons show differential neural responses to two exploratory contexts, supporting two conclusions assessed by the authors. (i) Coding of contexts is beyond the dichotomy of 'dominant' vs. 'exploratory' but contains more specific information regarding richer contextual content. However, critical evidence to distinguish between two 'dominant' contexts by ACC neurons needs to be provided. Moreover, whether the coding for exploratory contexts results from other factors, such as overall reward rate, needs to be clarified. (ii) Because ACC neurons differentiate two exploratory contexts in which reward is unavailable, the authors believe that reward cannot explain the observed neural correlates of strategy prevalence, which is an over-generalized statement since the 'dominant' and 'exploratory' contexts are different in reward availability. Besides, the authors' claim, used to support the irrelevance of reward expectation, that expert rats will likely sample the 'exploratory' sequence instances without any immediate reward expectation lacks clear evidence.

6) I find the manuscript often hard to follow primarily because of the usage of many jargons and extended expressions and sentences, which are unnecessary and could be replaced with more plain language to improve readability for a broader audience. To list a few:

i) behavioral framework = behavioral task?

ii) searching a structured space of action sequences = nose poking?

iv) restructuring of ACC network configuration = neural activity change in ACC?

v) persists through a substantial ensemble reorganization = persistent neural code?

vi) tags ACC representations with contextual content = ACC encodes context?

vii) organizing principles for the ACC ensemble dynamics = how ACC encodes information?

viii) representations of individual strategies are also marked with contextual content = encoding of strategy per se?

Besides, the introduction needs to be narrower in scope, especially at the end, to successfully frame and specify the questions the current study would like to answer. The last paragraph in the introduction contains a lengthy summary of the results. But since the task and results are so complicated, it is almost impossible to understand it without going through the following results in detail. Also, throughout the manuscript, many statements are too broad or vague to some extent. There is room to improve the introduction to set a better ground for the current study and other parts of the manuscript for a broader audience.

*Reviewer #3 (Recommendations for the authors):*

(1) I found the claims of the manuscript difficult to assess because many basic features are not described or shown in graphical form, and the abstracted analytical methodology (and language) is often difficult to understand. For instance, the description of behavior is incomplete relative to the description of the task in the methods. Rats learned many sequences, but the results of only one are shown. It is important to show whether the rats had biases for particular sequences, particularly for the interpretation of neural data. Otherwise, neurons might be revealing preferred vs non-preferred strategies. Showing the entropy of response strategies (i.e. sequences parsed into triplets and measuring the resulting entropy of triplets) may be a compact way to do so. In general, I recommend reorganizing the manuscript in a more incremental and linear manner.

(2) The primary argument provided that neurons encode strategy (Figure 1e) is based on the comparison to L in the RLL rule vs LLR. But a comparison is made between the 2nd element of the former rule and the first of the latter one. There is clearly a potential for confound by reward expectation. Even though the reward was withheld on the analyzed trials, there is likely different reward expectation. Specifically, the L response in the first position follows a negative reward prediction error. This analysis would be much more compelling by controlling for position and reward. Why not analyze the response to the 2nd element in the sequences RLL and RLR? This way, they are both in the same position and have the same local context (follow R). Reward expectancy is much less of a nuisance variable in this scenario.

(3) I am not convinced that it is valid to compare the 3 sequence responses to single responses in the 'competitor task' (Figure 1.d). The reward expectation is much different because there is a non-zero probability of reward after every choice. Furthermore, readers need to know much more information. Are these different animals with distinct training regiments, differences in the apparatus, etc.? Presumably, rats are going to the feeder after responses in the competitor task more often than the sequence task. I did not find in the present manuscript where the feeder was located relative to the response ports.

(4) My primary critique of the paper is that alternate explanations for variance are not sufficiently tested. Presently, this is done very piecemeal and is scattered throughout the results. The authors do address some of the possible confounders, but not to a level of sufficient rigor to support the overall claim that it is a dominant/exploratory strategy that accounts for differences in neural firing. In no particular order, I think the primary confounders are: reward expectancy; response vigor; posture during nose poke (which may depend on prior choice), position in sequence; side chosen, a relative time during the session (motivation and/or typical run-down of neural firing during long operant tasks), and rule certainty.

The potential for reward confound is fairly obvious. The authors should use some model of reward expectation as an estimate of reward expectation.

The authors need to do more to negate the possibility that neural variance reflects features of movement and posture. The authors mention an analysis visually presented in supplementary data. It is not clear that the principle component of trajectory is the best predictor. I recommend searching the feature space in an attempt to account for the most variance. For instance, body posture, head angle, and maximum velocity may provide superior predictive power. I am particularly concerned because the exploratory sequences appear to be often detected because of slowing in at least some phases of the responses. It is not clear whether to expect a linear relationship between motor output and neural signalling. A link function may be necessary.

Neural activity often drifts over the course of long sessions in which animals perform many trials of an operant task. The authors need to dismiss that this is a possibility, particularly in their analysis of dominant vs exploratory strategy. It is not evident from the manuscript how many trials animals are doing and how long the blocks are. Because of the number of possible sequences, it seems likely that the time between each strategy being the dominant versus exploratory may allow for task-independent drift (fatigue or mechanical instability) in neural activity. Authors should show that difference in firing for dominant/exploratory strategy is not influenced by drift in neural firing over time. If it is, this variance must be parsed out.

(5) Figure 1B: it is not clear why we are shown only one response sequence. Why not show performance accuracy overall strategies? The legend indicates that this is 'across the behavioural dataset'. Does this mean all sessions were concatenated? The sampling is unbalanced among animals. It is important to know if they performed similarly, so showing the mean of each animal is useful.

(6) I recommend tempering the language in the abstract, which mentions intelligence and complex settings. The relation of the present work to these topics is better presented in the discussion if the authors think them relevant.

(7) The firing rates shown in Figure 1 and 4 appear quite high for rat pyramidal neurons in mPFC. Are these cells typical of the dataset, or outliers in terms of firing rate? A quantification of the general activity over the population of cells is needed. Moreover, a plot of firing rate versus spike width would provide valuable insight into the distribution of putative pyramidal and interneurons.

---

## [Author Response]

**Strategy-specific action encoding**

Reviewer 1 writes: “The authors provide strong data indicating that a given L or R response is associated with distinct ACC activity depending on which sequence that response is embedded within, a finding reminiscent of other reports in multiple brain regions. While not a criticism per se, I was interested in the center port responses, also embedded within unique sequences, yet never preceding reward. A key difference in the performance of a given R or L response is that it is sometimes the terminal response, and thus the rat knows a given R or L response to be sometimes reinforced in one of the contexts, but not the other, in each of these comparisons. I wonder if there was an opportunity to cleanly demonstrate the context dependence of a given individual action by comparing center port responses across distinct sequences.”Reviewer 2 writes: “The authors examine neural responses to 'R' in LLR vs. RRL. But apparently, the two Rs are in different steps (step 3 in LLR and step 1 or 2 in RRL) within the two three-step sequences. Steps thus might explain such differential responses to the R in different sequences. Many neurons would likely show selectivity to different steps regardless of specific actions (L or R), which need further testing.Reviewer 3 writes: “the authors compare the encoding of one action (L) in two sequences (RLL and LLR). However, the analyzed action occurs in different local contexts. In the first, it is the middle action, and in the second it is the first action following a reward omission. Even though the reward is withheld, the rat presumably has some reward expectation….“…The primary argument provided that neurons encode strategy (Figure 1e) is based on the comparison to L in the RLL rule vs LLR. But a comparison is made between the secondnd element of the former rule and the first of the latter one. There is clearly a potential for confound by reward expectation. Even though the reward was withheld on the analyzed trials, there is likely different reward expectation. Specifically, the L response in the first position follows a negative reward prediction error. This analysis would be much more compelling by controlling for position and reward. Why not analyze the response to the secondnd element in the sequences RLL and RLR? This way, they are both in the same position and have the same local context (follow R). Reward expectancy is much less of a nuisance variable in this scenario.”

We have now expanded our argument about strategy encoding to address the Reviewers’ concerns, incorporating their insightful suggestions in two distinct ways:

1) Most importantly, we implemented a variant of the last suggestion from Reviewer 3. We have avoided using ‘RLR’ sequence in our tasks because it can be hacked quite effectively through basic alternation- an innate bias many naive rodents display. Nevertheless, we loved the spirit of the suggestion: match the immediate past choice as well as the distance to reward. We therefore took advantage of the sessions where our animals were tasked with finding all four non-trivial 4 letter sequences to demonstrate robust decodability of the first ‘R’ in ‘RRL’ vs ‘RLL’ (or the first ‘L’ in ‘LLR’ vs ‘LRR’) (new Figure 1d-e, last condition).

2) We have also established that all conclusions of the series of decoding analyses remain valid even when the decoding window is shifted to the center port entry – a common step at every trial initiation (new Figure 1d-e). Indeed, the original set of decoding analyses, done using a decoding window anchored on choice port entry has now been moved to the supplementary materials.

**‘Dominant’/’exploratory’ vs ‘rewarded’/’unrewarded’**

Reviewer 2 writes: “(ii) Because ACC neurons differentiate two exploratory contexts in which reward is unavailable, the authors believe that reward cannot explain the observed neural correlates of strategy prevalence, which is an over-generalized statement since the 'dominant' and 'exploratory' contexts are different in reward availability. Besides, the authors' claim, used to support the irrelevance of reward expectation, that expert rats will likely sample the 'exploratory' sequence instances without any immediate reward expectation lacks clear evidence.”Reviewer 2 also writes: “(5) In Figure 3e, data from a small subset of sessions show that neurons show differential neural responses to two exploratory contexts, supporting two conclusions assessed by the authors. (i) Coding of contexts is beyond the dichotomy of 'dominant' vs. 'exploratory' but contains more specific information regarding richer contextual content. However, critical evidence to distinguish between two 'dominant' contexts by ACC neurons needs to be provided. Moreover, whether the coding for exploratory contexts results from other factors, such as overall reward rate, needs to be clarified.”Reviewer 3 writes: “(4) …The authors do address some of the possible confounders, but not to a level of sufficient rigor to support the overall claim that it is a dominant/exploratory strategy that accounts for differences in neural firing. …I think the primary confounders are: reward expectancy…”

We apologize for not spelling out the logic of this argument better. Our reasoning was as follows: in a small subset of sessions, we included both ‘LLR’ AND ‘LLLR’ blocks (for this analysis, we were sure to pick sessions, in which proficiency with ‘LLLR’ was achieved). This permitted us to look at exploratory ‘RRL’ sequence instances (never rewarded) in these two distinct conditions. Note that by our selection filter, for any sequence instance to be even considered for the ‘exploratory’ dataset, the block-specific target (‘LLR’ and ‘LLLR’ respectively in this experiment) has to have been discovered by the animal, and moreover, it has to have come to dominate the animal’s choices. Thus, in the old Figure 3e (new Figure 3d), we were comparing ACC representations of ‘RRLs’ done as deviations from persisting with a previously discovered, rewarded ‘LLR’ or a previously discovered, rewarded ‘LLLR’. Both subsets of RRLs were exploratory, and thus associated with the same reward unavailability, yet as the old Figure 3e (new Figure 3d) shows, the associated global ACC neural states were as different as those between ‘exploratory’ and ‘dominant’ ‘RRL’s. This led us to conclude that (1) ‘rewarded’/’unrewarded’ is insufficient to capture the difference in global states, and (2) ‘exploratory’ is insufficient as well.

To answer Reviewer 2’s comment: “However, critical evidence to distinguish between two 'dominant' contexts by ACC neurons needs to be provided”:

The differential encoding of ‘RRL’ between ‘LLR’ and ‘LLLR’ contexts is in itself an indication that ACC neurons can tell the difference between the two contexts. Nevertheless, below we provide independent evidence to this effect by demonstrating that we can decode whether an ‘R’ at the end of a ‘(L)LLR’ originates in the ‘LLR’ block or in the ‘LLLR’ block.

More generally, it is still, however, a fair point that while our expert animals likely build on their extensive experience to expect only the target sequence to secure reward, some possibility remains that a transient one/two- instance exploratory resurgence of ‘RRL’ might come with some non-zero reward expectation (for instance, if animals were thereby pre-emptively evaluating the possibility of a block transition).

It is also fair that at least in some ‘dominant tails’ – stretches of determined commitment to the previously rewarded dominant sequence past unsignalled block transitions – our animals simply hadn’t noticed the transition yet, and thus hadn’t changed their reward expectation. We therefore sought to develop an additional setting aimed at disambiguating ‘reward’/’no-reward’ classification of the global ensemble state from ‘dominant’/’exploratory’ (or some more nuanced variation of the latter).

We built on one parsimonious interpretation of the distinct global ensemble states evident from the differential encoding of exploratory ‘RRL’ in ‘LLR’ vs ‘LLLR’ contexts (and other findings at the level of global contextual encoding). If the global contextual state of the ACC ensemble represents the inferred target sequence, then we should expect that state to be similar BOTH when the dominant sequence in the animal’s behavior is indeed aligned with the latent rewarded target, and when it is not, i.e. when the animal perseverates with a non-rewarded sequence at the expense of others.

In a new set of animals, we have now collected additional data in sessions where the target sequence set included not only the familiar ‘LLR’ and ‘RRL’, but also the recently introduced ‘LLLR’ and ‘RRRL’. We discovered, that under such conditions of early ‘LLLR’/’RRRL’ acquisition, animals’ choices in the ‘LLLR’ and ‘RRRL’ blocks were often dominated by the cognate familiar shorter sequence due to a lack of proficiency with the longer targets (see example in Figure 3f).

We now demonstrate that in such cases, ensemble states associated with such dominant but unrewarded ‘LLR’ sequence instances also do not cluster far away from ones observed earlier in the same session when the dominant ‘LLR’ strategy matched the rewarded target (new Figure 3f). Furthermore, since persistence with an unrewarded sequence in this case occurred after the animals had detected an unsignalled block change and switched away from the previous strategy (note the switch in new Figure 3f, middle epoch), the similarity of ensemble states cannot be explained by a lack of rule switch awareness that potentially confounds the dominant ‘tails’ example. Together with the evidence we explored in the original submission, these data argue that ‘rewarded’/’unrewarded’ dichotomy does not provide a parsimonious classification of the discrete global ACC ensemble states.

**Exploratory sequence instances vs errors of execution**

Reviewer 2 writes: “(2) To identify exploratory sequence instances in the behavioral data, the authors try to remove many action sequence instances that could be explained by factors other than 'exploration.' While this effort is appreciated, it remains to be answered whether one could pinpoint the actual 'exploratory' instances or their existence. To do so, we need to figure out if a mistake made by rats is random, based on a false belief, or only caused by disengagement. Even if there are some exploratory sequence instances in the behavior, other causes might contaminate identified instances, which could promote the current findings on the coding of strategy prevalence. For instance, the authors use a criterion combining the within-side duration and side-to-center duration in step 3 to remove overlapped and long-break sequences. It, however, can only partially rule out other possible contamination as many trials display long choice duration even in step 1 (Figure S3), and the variability is also substantial. One suggestion is that the authors try a more stringent criterion and test whether the primary findings hold firmly.”

There seem to be two separate, although related questions at the core of this concern:

i) Is there even evidence that *any* of the deviations from the target sequence represent anything other than errors of execution?

ii) How robust are the described effects to progressive culling of the set of all putative exploratory sequence instances?

i) Two observations argue against deviations from the discovered target sequence being mere errors of execution:

– Many of the deviations contain direct concatenations of several instances of an alternative sequence (see, for example, Figure 2b). Moreover, as can be inferred from Figure 2c, most exploratory sequence instances occur in bouts (otherwise, for any given exploratory sequence instances, the local prevalence of that sequence in recent past history would be close to zero, and the histogram would be much more squished to the left (see example of such a distribution in the second argument below))

– The type of bout-based putative exploratory resurgence of specific alternative sequences captured in Figure 2c is something we observe in animals that have experienced those sequences as latent targets. Below we give two examples:

– Across-animal comparison. Imagine that all putative instances of, say, ‘LLR’ we observe arose as mere errors of execution, despite the apparent correlation of co-occurrence (Figure 2c) suggesting that the exploratory sequence instances occur in bouts. Then, if we were to examine the dataset from animals trained on a distinct set of sequences – one that never included ‘LLR’ as a latent target – we would expect to encounter ‘LLR’s with bout frequency that is on par with what we see in the dataset in this manuscript. Contrary to that prediction, the distribution of local ‘LLR’ sequence prevalence look dramatically different in animals trained on ‘LRLRR’ and ‘LRLRLRR’ sequences (new Figure 1—figure supplement 2a-b).

– Within-animal comparison. For animals that first become proficient with ‘LLR’ and ‘RRL’ sequences, and later learn the longer ‘LLLR’ and ‘RRRL’ sequences to expert level, we see a noticeable increase in the incidence of the longer sequences in deviations from the dominant target after that learning stage­ (new Figure 1—figure supplement 2c-d).

We now mention this explicitly on p.6 of the manuscript:

“…Nevertheless, clear deviations from this dominant pattern, with the animals’ choices appearing instead to conform to other possible target sequences, were also present within all blocks (Figures. 1c, 2a-c). Although some of the deviations from the currently rewarded target sequence may represent errors of execution, the presence of bouts and even direct concatenations of previously reinforced sequences (Figure 1 —figure supplement 2a-d) argue that at least some of these deviations represent transient exploratory resurgence of alternative sequences.”

ii) The second question is something we worried about from the beginning, when we first observed the phenomena that are the subject of this study, at that time on the full set of putative exploratory sequence instances. For the paper, we had settled on the harshest set of criteria that still left enough sequence instances to give the dataset statistical power. Nevertheless, it is important to emphasize that we had verified the robustness of the findings to the inclusion of progressively harsher selection criteria at each sub-selection step along the way.

**Activity modulation by strategy prevalence vs by reward or certainty**

Reviewer 1 writes: **“**I find it difficult to disentangle prevalence encoding and impacts of reward in the way the data and interpretation are presented in some areas of the text. While neural encoding may not reflect trial-by-trial reward receipt, clearly the rat's decision to repeat a given sequence or initiate a new sequence is impacted by reinforcement parameters and reward expectation. Thus being very exact in the interpretation would be helpful.”Reviewer 2 writes: “The authors have tried to remove the influence of reward expectation by only looking at different exploratory states since no reward is available in these trials. But reward expectation can still explain differential firing to the dominant vs. exploratory strategies. The dominant tail effect seen in Figure 3f following the change of rules does not help because reward expectation, like strategy prevalence, also follows the subjective belief of the rats but not the actual task rule…”Reviewer 3 writes: **“**I think the primary confounders are: reward expectancy… and rule certainty…The potential for reward confound is fairly obvious. The authors should use some model of reward expectation as an estimate of reward expectation.”

In response to the Reviewers’ requests, we have extended our analysis and narrative related reward in the following 3 ways:

1. We have expanded the set of contexts where a specific behavioral sequence was performed under a ‘no-reward’ condition. Specifically, we have now collected new data in a setting where persistent, but unrewarded ‘LLR’ and ‘RRL’ sequence instances dominated within the unfamiliar ‘LLLR’ and ‘RRRL’ contexts. What makes this fundamentally different from the case of ‘dominant tails’ is that the observed persistence with an unrewarded sequence in this case occurred after the animals had detected an unsignalled block change and switched away from the previous strategy (Figure 3f) and thus cannot be explained by a lack of rule switch awareness.

The addition of this extra 'no reward’ condition permitted us to more convincingly demonstrate (a) that the global ‘dominant’ ACC ensemble state relates to strategy dominance rather than the associated reward (see section ‘Dominant’/’exploratory’ vs ‘rewarded’/’unrewarded*’* above and new Figure 3f ), and (b) that strategy prevalence makes at least some reward independent contribution to modulating ACC activity during the execution of a specific strategy (Figure 4 —figure supplement 2b-c). We also note that since rule certainly captures one’s belief about strategy outcome, observing activity modulation by strategy prevalence under conditions of long-term persistence with an unrewarded strategy argues against that modulation reflecting rule certainty.

2. We took advantage of the fact that in our dataset, the sequence prevalence parameter and the reward prevalence parameter are sufficiently decorrelated to probe – using multiple linear regression – whether the unexpected robustness of the simple strategy prevalence model above in explaining a marked fraction of ACC neural activity variance during the execution of a specific sequence derives mostly from the impact of successfully procured reward. By separately considering an expanded model with an ‘overall reward prevalence’ term and one with a ‘sequence-specific reward prevalence’ term, we demonstrate that the tracking of reward by the ACC ensemble is strategy specific – indirectly tying that to strategy prevalence – but that strategy-specific reward prevalence alone was insufficient to explain the activity modulation during strategy execution (new Figure 4d).

3. We have re-organized the narrative, expanding the discussion of these issues into a separate section (see section The tracking of reward by the ACC neural ensemble is strategy-specific, however reward prevalence is insufficient to account for activity modulation on p.17 of the revision and below):

The tracking of reward by the ACC neural ensemble is strategy-specific, however reward prevalence is insufficient to account for activity modulation

“Models of decision-making rarely include a summary statistic of the agent’s recent behavioral choices – such as the local strategy prevalence that we examine here. This prompted us to next consider how the observed modulation of ACC ensemble activity by local strategy prevalence might interact with any modulation exerted by the external reward – a more commonly considered parameter directly related to the concept of valuation thought to engage circuit computations in the ACC (Boorman et al., 2009; Kennerley et al., 2009; Kolling et al., 2012; Luk and Wallis, 2013). Indeed, variation in strategy prevalence is necessarily accompanied by variation in detailed reward history and thus understanding the interplay between the two in shaping ACC neural dynamics may shed further light on how the animals parse their environment.

We first established that local strategy prevalence provides at least some modulatory influence independent of external reward by evaluating the performance of the prevalence model trained exclusively on sequence instances executed under ‘no reward’ conditions. Specifically, we evaluated two distinct cases: ‘exploratory’ sequence instances in sessions comprising blocks of the familiar ‘LLR’ and ‘RRL’ targets, and persistent ‘LLR’ and ‘RRL’ sequence instances within the unfamiliar ‘LLLR’ and ‘RRRL’ contexts. The observed model performance in both cases was significantly more robust than what would be expected if most of the observed activity variance during sequence execution arose from the statistics of the associated reward (Figure 4 —figure supplement 2). In further support of the conclusion that sequence prevalence shapes ACC neural dynamics independent of the associated reward, we observed little activity modulation for sequence instances within ‘dominant’ tails – a period when sequence prevalence remains high, but reward expectation should rapidly diminish (Figure 3f). Combined, these data argue that a summary statistic of that strategy in past choices exerts a continuous modulation of the strategy representation in ACC and raise the question of how this modulation interacts with any shaping of the ACC activity by the external reward.

ACC is known to both multiplex information about reward and individual actions (Hayden and Platt, 2010), as well as track progression to reward over multiple steps of an action sequence (Shidara and Richmond, 2002). To determine whether the unexpected robustness of the simple strategy prevalence model above in explaining a marked fraction of ACC neural activity variance during the execution of a specific sequence derives mostly from the impact of successfully procured reward, we incorporated an additional ‘reward prevalence’ term into the linear model and determined the relative contribution of the ‘sequence prevalence’ and ‘reward prevalence’ terms to the expanded model’s performance (see below). Provided that the two parameters are sufficiently decorrelated in our dataset to avoid collinearity – due to factors like reward omission, exploratory sequence instances and transient persistence with off-target sequences– their weights in the expanded model would reflect the relative contribution to the model’s performance from the summary behavioral statistic itself and from the recently procured reward. To account for the possibility that reward statistics may be processed by, and exert influence on, the ACC circuitry at either the single trial- or the single sequence level, we separately considered a model with an ‘overall reward prevalence’ term and one with a ‘sequence-specific reward prevalence’ term. As such, a reward obtained for the target ‘RRL’ sequence before an exploratory, off-target, ‘LLR’ sequence instance would contribute to the ‘overall reward prevalence’ term, but not to the ‘sequence-specific reward prevalence’ term in models aimed at explaining neural activity variance during ‘LLR’ execution. We computed each ‘reward prevalence’ term by weighing the relevant previous rewards with the same temporal filter used to calculate sequence prevalence and identified a subset of behavioral sessions for which the ‘sequence prevalence’ and each of the ‘reward prevalence’ parameters were sufficiently decorrelated in our dataset (Methods).

Does the observed marked modulation of ACC neural dynamics during different instances of sequence execution outside of some particularly persistent ‘no reward’ contexts arise largely from variation in recent reward history? To resolve this, we determined whether the model’s weight for the ‘sequence prevalence’ term became negligible once the ‘reward prevalence’ term was added to the model. Contrary to this expectation, neither the ‘overall reward prevalence’ nor the ‘sequence-specific reward prevalence’ terms dwarfed the contribution from the ‘strategy prevalence’ term to the model’s performance, with the ‘overall reward prevalence’ making a particularly small contribution to the model’s explanatory power (Figure 4d). Furthermore, while the contribution from the ‘sequence-specific reward prevalence’ was on par with that of ‘strategy prevalence’, it was clearly insufficient on its own to account for the model’s explanatory power (Figure 4d). Combined, these observations argue that the tracking of reward by ACC is strategy-specific, and that both a summary statistic of the specific self-guided behavioral strategy in recent past and a statistic of the associated reward shape the ACC neural dynamics during strategy execution, further highlighting the central role of the animal’s strategy choice rather than the external rule in this process.”

**Modulation by strategy prevalence vs by movement-related parameters**

Reviewer 1 writes: “Would it make sense to focus the control analyses (vigor, reward history, and so on) on those sessions/ensembles with greater variance explained, ie, perhaps there might be greater sensitivity to detecting interactions among variables within ensembles recorded more rostrally?”Reviewer 3 writes: “The paper would benefit from analyses, such as multiple regression over all possible predictive variables, to evaluate the relative amount of neural signal variance attributable to strategy dominance compared to other information….

We have now refined our evaluation of the possibility that the observed linear modulation of ACC activity in fact reflects modulation by co-varying movement-related parameters by combining these two insightful suggestions. Specifically, we have expanded the section devoted to this issue (p. 16) where we focused in more detail on the dataset recorded in the more rostral part of the cingulate cortex and asked whether adding a separate movement-related parameter to the linear model would shift the explanatory power away from the sequence prevalence term. We report that the expanded models revealed only a minor contribution of the execution vigor or of the specific trajectory to the overall model’s performance – an observation consistent with the relatively poor performance of the relevant single parameter models included in the original submission. We write:

“The comparatively low amount of neural activity variance in the motor regions M2 and SMC explained by strategy prevalence suggests that the observed modulation is unlikely to reflect movement parameters that may co-vary with changes in the animal’s dedication to a particular strategy. To establish this more explicitly, we directly quantified trial-by-trial measures of movement vigor (sequence execution time) and kinematics (the first principal component of movement trajectory) for each instance of sequence execution and then asked how well linear models that incorporate these parameters performed at explaining variance in ACC firing rates (Methods). We first evaluated the performance of an equivalent linear model that used either execution time or the first principal component of movement trajectory as a single parameter; neither model could explain ACC neural activity variance as well as sequence prevalence (Figure 4 —figure supplement 3a). We then focused in more detail on the dataset recorded in the anterior part of the cingulate cortex that displayed the strongest modulation by strategy prevalence (Figure 4b,c), and asked whether adding a separate movement-related parameter to the linear model would dwarf the explanatory power of the prevalence term (Methods). To ensure that the resulting weights on the two parameters would directly reflect their relative contributions, we z-score normalized all the variables to 0 mean and standard deviation of 1 prior to fitting the model. Consistent with the relatively poor performance of the single movement-related parameter models, the expanded models revealed only a minor contribution from those motor parameters to the overall model’s performance (Figure 4 —figure supplement 3b). While it remains possible that more nuanced aspects of movement, like the animal’s posture, contribute to the observed activity modulation, these observations suggest that gross movement-related parameters are not behind the explanatory power of strategy prevalence.”

Reviewer 3 also writes: “It is not clear that the principle component of trajectory is the best predictor. I recommend searching the feature space in an attempt to account for the most variance. For instance, body posture, head angle, and maximum velocity may provide superior predictive power. I am particularly concerned because the exploratory sequences appear to be often detected because of slowing in at least some phases of the responses.”

Unfortunately, the specifics of the video data that accompany this dataset precludes us from doing more sophisticated pose tracking. To highlight the limitation of our analysis, we included the following comment to the relevant text section:

“While it remains possible that more nuanced aspects of movement, like the animal’s posture, contribute to the observed activity modulation, these observations suggest that gross movement-related parameters are not behind the explanatory power of strategy prevalence. “

It is worth mentioning, however, that the momentary slowing down that accompanies the demarcation by an animal of a completed sequence happens outside of the temporal window used in all analyses in this manuscript.

Finally, Reviewer 3 notes: “It is not clear whether to expect a linear relationship between motor output and neural signalling. A link function may be necessary.”

We regret that the compact wording in the original submission left the logic behind out analyses unclear. The specific question we aimed to address was whether the linear component of the activity modulation we were attributing to sequence prevalence was instead the result of the underlying variability in motor parameters. As such, we selectively evaluated the explanatory power of a linear relationship. We hope that the expanded explanation, along with the inclusion of the multiple linear regression analysis, clarifies the logic of the argument.

Additional details about behavior is needed: accuracy on all strategies, how many blocks per session, did rats have individual biases in strategy, etc.

Reviewer 1 writes: “The main text provides an intriguing narrative but lacked very many of the quantitative details one would like to evaluate the claims, requiring a lot of back and forth between the results and the methods, and the figure legends. The number of rats, the number of sequence occurrences, the number of neurons simultaneously recorded, etc. For any given analysis of a set of sessions, the reader is not told if the sessions switched among 2, 3, or 4 sequences, nor what the reinforcement parameters were, all variables that potentially impact the behavior and neural encoding, and therefore the decoding analyses.”

We apologize for the confusion that arose from our laconic style in the original submission; the importance of stating this clearly is further underscored by the fact that in the revised version, we now deviate on the target sequence set specifically for the decoding analyses (see ‘Strategy-specific action encoding’ section of the response above).

Initially, we restricted our analysis of neural activity to sessions with ‘LLR’/’RRL” transitions (with the exception of a single control for the encoding of ‘exploratory’ sequence instances in two distinct ‘dominant’ contexts). We apologize for not making the rational here clear: this choice ensured that we removed additional uncertainty inherent in parsing circularly permuted sequences when identifying putative exploratory sequence instances (e.g. disambiguating a putative ‘LRR’ from a putative ‘RRL’ in ‘LRRL’). In revision, we both outline this argument more clearly. Furthermore, when exceptions to this general rule are made, we not only mention it, but explain the purpose and the task design.

Specifically, to explain the rationale for largely focusing on ‘LLR’/’RRL’ sessions when analyzing contextual modulation of sequence representation, we now include the following statement in the text (p. 9):

“In contrast, parsing is much harder for the deviations from the dominant target outside of clear sequence concatenations. This is particularly challenging in cases where the task set contains circularly permuted strategies: does a ‘LRRL’ deviation during a ‘LLR’ block reflect a ‘LRR’ instance, a ‘RRL’ instance, or neither? And while the additional uncertainty inherent in parsing circularly permuted sequences can be resolved by focusing on sessions that contained only ‘LLR’ and ‘RRL’ blocks, not even every apposition of ‘Left’, ‘Left’ and ‘Right’ choices in a ‘RRL’ block will reflect an exploratory instance of the ‘LLR’ sequence. We therefore next sought to delineate an objective criterion for including any lone putative exploratory sequence instance in the subsequent investigation; with the exception of a few pre-specified control experiments, the remaining analyses are restricted to ‘LLR’/’RRL’ block switches. *“*

Once key place where we deviate from focusing on ‘LLR’/’RRL’ sessions is in the decoding section. Here, in response to Reviewers’ suggestions, we now also take advantage of sessions that tasked our animals to find all four non-trivial 3 step targets (see more on that below in the Strategy-specific action encoding section of our response to the “The most significant weakness” part of Essential Revisions). For these specific analyses, we could avoid the uncertainty of parsing by restrictoing the decoding analyses to those sequence instances that both matched the latent rewarded target and had established dominance in the animal’s choices. Indeed, analyzing such dominant sequence instances for each target sequence was sufficient to answer the question of whether action representations in ACC is strategy-dependent.

We now write in the opening portion of that part of the manuscript (p.6):

*“*… we first verified that the specific choice of strategy in this more complex sequence task is reflected in ACC neural dynamics by establishing that individual sequences could be decoded from ensemble activity. For these strategy decoding analyses we focused on those periods when each sequence of interest both matched the latent rewarded target and had established dominance in the animal’s choices.”

We also made sure to specify when the inclusion of these sessions was necessary. We now write (p.7):

“The observed differences in ACC representations for a given ‘L’ (or ‘R’) action within the sequences ‘LLR’ and ‘RRL’ could reflect the differential encoding of distinct strategies that are currently being pursued. However, the ‘L’ (or ‘R’s) in these different sequences are also associated with differences in the immediate history of other actions (‘L’ in ‘RRL’ follows an ‘R’, while one of the ‘L’s in ‘LLR’ follows an ‘L’) and the proximity of rewards (no ‘R’ in ‘RRL’ is ever rewarded, whereas most ‘R’s in ‘LLR’ are). We therefore carried out additional analyses to ask whether either of these factors – surrounding actions, or reward proximity – could account for the apparent strategy encoding in ACC, taking advantage of a subset of sessions that tasked our animals with discovering additional 3 letter target sequences.”

As for the reinforcement parameters, although the fraction of reward omission varied across our dataset, it was always matched across different target sequences within a session. Thus, reinforcement parameters did not come into play for the decoding analyses.

Reviewer 2 writes: “(1) The authors claim the performance of the rats to be 'robust.' Is there an objective measurement to test such robustness? The conditioned probabilities in Figure 1b might provide some evidence. But it is not straightforward to comprehend what a probability of 0.4 means relating to behavioral performance. It would be helpful to have a learning curve showing baseline behavioral performance and whether expert rats have reached their behavioral asymptotes.”

We apologize for not referencing the figures behind this statement. In our minds, 3 figures play into that claim- Figure 1b, Figure 2c and new Figure 1—figure supplement 1.

One straightforward way to define performance is the probability (or prevalence) that the animal is doing the target sequence. In early experiments, we found that the animals’ performance indeed asymptotes in a version of the task where only one target sequence is used over the course of many sessions. However, in the dynamic version of the task used in this manuscript – one with repeated switches between blocks characterized by different target sequences – such performance can locally vary within a session even for expert animals because they repeatedly search for a new target sequence or deviate to explore the alternatives. Thus, we feel that the distribution of local sequence prevalence values for dominant blocks (Figure 2c) is more informative than a learning curve. In essence, that analysis characterizes the likelihood for any given dominant ‘LLR’ (or ‘RRL’) that the animal has locally dedicated a certain amount of attention to that target sequence (with ‘1’ signifying that the animal has not deviated from the target sequence over the past 20 trials). What Figure 2c fails to show is that the target is often cleanly concatenated even when transient deviations from that dominant strategy crop up over the course of the 20-trial window; that information is instead presented in Figure 1b.

Finally, the ability of our animals to detect and adjust to block transitions within tens of trials (old Supplementary Figure 1, new Figure 1—figure supplement 1a) is another aspect of robust task performance we would strive to highlight. We have made sure to reference these 3 Figures in the context of the robustness claim.

Reviewer 3 writes: “(1) …the description of behavior is incomplete relative to the description of the task in the methods. Rats learned many sequences, but the results of only one are shown. It is important to show whether the rats had biases for particular sequences, particularly for the interpretation of neural data. Otherwise, neurons might be revealing preferred vs non-preferred strategies. Showing the entropy of response strategies (i.e. sequences parsed into triplets and measuring the resulting entropy of triplets) may be a compact way to do so.”

We apologize for two separate sources of confusion here.

The first source stems from our misguided attempt to simplify axis labels. Our summary plots always show data pooled across ‘LLR’ and ‘RRL’ strategies, which we think of as interchangeable. Thus, for instance, what was labeled in the original Figure 1b as p(LLR|LLR) is, in fact, p(LLR|LLR in ‘LLR’ block) OR p(RRL|RRL in ‘RRL’ block). We have now expanded the relevant labels to reflect the fact that both sequences contributed to the analyses.

Most importantly, however, all analyses about activity modulation by local and global context are always done *for different instances of the same strategy* (ie action sequence), and only during epochs when the target sequence has established dominance. For instance, whereas we compare exploratory ‘RRL’ to dominant ‘RRL’, or exploratory ‘LLR’ to dominant ‘LLR’, we never compare ‘RRL’ to ‘LLR’. Thus, preference to a sequence doesn’t account for any spotted difference in such analyses.

Thus, putative preferences for a difference strategy would not factor into the interpretation of neural data. We have now emphasized this point several times throughout the revision to help the reader. For example, we conclude the opening section on strategy encoding on p. 8 by saying:

“Combined, these observations argue that individual multi-step sequential strategies have distinct ACC representations, permitting us to next evaluate whether the ACC representation of any specific sequence is further modulated by the specific context in which a particular instance of that sequence is being executed. “

Reviewer 3 also writes: “(5) Figure 1B: it is not clear why we are shown only one response sequence. Why not show performance accuracy overall strategies? The legend indicates that this is 'across the behavioural dataset'. Does this mean all sessions were concatenated? The sampling is unbalanced among animals. It is important to know if they performed similarly, so showing the mean of each animal is useful

We again apologize for the confusion that arose from our attempt to keep the axis labels simple. In all behavioral analyses, the data was always pooled for the ‘LLR’ and ‘RRL’ sequences (they are, in our minds, equivalent). We have now corrected the labels (e.g. replacing p(LLRLLR)|p(LLR) with p(LLRLLR)|p(LLR) or p(RRLRRL)|p(RRL)).

It is again important to emphasize that all analyses about activity modulation by local and global context are always done *for different instances of the same strategy* (ie action sequence), and only during epochs when the target sequence has established dominance. Thus, a somewhat better performance on one strategy over another – if present – would not factor into the interpretation of neural data.

**Reinforcement effects not sufficiently integrated/discussed**

Reviewer 1 writes: “Some greater attention to the behavioral parameters could be helpful, especially regarding the impact of reward rate on behavior. For example, looking at some of the figures of individual rat behavior, exploratory sequences seemed triggered by reward omission. Is this just a chance for the examples chosen or is there something systematic here? Upon block switch, how exactly does the switch in sequences emitted by the rat track with reinforcement history? The authors mention that reinforcement probability differed across sessions, and one would thus expect switching behavior would as well. Because of the interesting existence of sometimes quite long 'tails' of performance of the original sequence after a block switch, I am wondering how the length of such tails relates to reinforcement rate parameters.”

Our original choice to give these details only partial attention stems from the fact that any potential differences in the rate, at which our animals adjust their behavior at block transitions would not affect the analysis or interpretation of the neural data for two reasons:

(1) With the exception of the strategy decoding section, all of the neural data analyses in the manuscript compare representations of the same given action sequence, just across different instances of its execution.

(2) Sequence instances selected always come from session epochs where one sequence has established a clear dominance in an animal’s behavior.

Nevertheless, we are happy to add the requested details below.

“…looking at some of the figures of individual rat behavior, exploratory sequences seemed triggered by reward omission. Is this just a chance for the examples chosen or is there something systematic here?”

This is a very insightful question, and one we could certainly devote more attention to in the manuscript. As we mentioned in the original submission, we reliably see exploratory instances of other sequences even in the absence of reward omission (i.e. in animals — like two in the included dataset — that have only ever experienced deterministic reward for where every instance of correctly executed target sequence) (new Figure 1 —figure supplement 2 f-g). Furthermore, even with animals that, through experience, always expect a certain degree of reward omission, we see examples of exploration that are not obviously triggered by reward omission (new Figure 1—figure supplement 2h).

That said, not unexpectedly, when evaluated across the entire dataset for animals trained under the conditions of periodic reward omission, there is indeed a spike in omission rate when aligned relative to the onset of exploratory bouts (new Figure 1—figure supplement 2i). This observation suggests that the onset of exploratory bout is indeed more likely to happen following an omitted reward of the dominant sequence.

We have now included this analysis in the revised version. Specifically, we write (p 6):

“Furthermore, while such transient deviations away from the dominant sequence were significantly more likely to follow the absence of an expected reward (Figure 1 —figure supplement 2i), similar strategy deviations were present even when no reward was omitted (Figure 1 —figure supplement 2f-h), suggesting that animals continue to sporadically sample other sequences even when not extrinsically prompted.”

“Upon block switch, how exactly does the switch in sequences emitted by the rat track with reinforcement history? The authors mention that reinforcement probability differed across sessions, and one would thus expect switching behavior would as well. Because of the interesting existence of sometimes quite long 'tails' of performance of the original sequence after a block switch, I am wondering how the length of such tails relates to reinforcement rate parameters.”

We find that at block transitions, neither the length of the ‘dominant tails’ nor the number of trials to commit to the new dominant strategy displays a significant dependence on reward omission rate (Figure 3—figure supplement 1). Consistent with this, we see “tails” even in animals that never experience reward omission. One possibility is that these “tails”, with their high sequence prevalence despite an utter lack of reinforcement represent a concerted effort by the animals to falsify the “this sequence is still valid” hypothesis.

**Too much unnecessary jargon reduces clarity**

Reviewer 2 writes: “I find the manuscript often hard to follow primarily because of the usage of many jargons and extended expressions and sentences, which are unnecessary and could be replaced with more plain language to improve readability for a broader audience….…Besides, the introduction needs to be narrower in scope, especially at the end, to successfully frame and specify the questions the current study would like to answer. The last paragraph in the introduction contains a lengthy summary of the results. But since the task and results are so complicated, it is almost impossible to understand it without going through the following results in detail.”

We apologize for the undue influence of run-on sentences more common in our native language. We thank the Reviewer for nudging us to make some changes, and for highlighting the difficulty with parsing the unnecessary detailed preamble of the original introduction. We hope that the simplified narrative (including the re-written concluding paragraph of the introduction, as well as many parts of the Results) will be easier to follow.

Reviewer 2 also raised a number of specific possibilities for changing the original wording, which we admittedly chose at times with the additional unpublished work from the lab in mind.

“(i) behavioral framework = behavioral task? “

We have implemented the suggested change.

(ii) searching a structured space of action sequences = nose poking?

In the end, we felt that “nose poking” does not quite capture the essence of our task design. However, we have simplified the wording:

“…requiring rats – in an apparatus that has ‘left’ and ‘right’ nose ports – to discover (without any explicit instruction) a specific rewarded sequence of ‘Left’/’Right’ choices from a larger set of structured possibilities.”

(iv) restructuring of ACC network configuration = neural activity change in ACC?

Since both the marked, abrupt, ensemble-wide changes in activity that accompany inference of block transitions and the more gradual modulation by local sequence prevalence are neural activity changes, we feel that “restructuring of ACC network configuration” is a more accurate, if somewhat mechanistic, account of the former.

(v) persists through a substantial ensemble reorganization = persistent neural code?

We worry that “persistent neural code” might suggest to a reader a stable neural representation. Instead, we are trying to convey that changes of neural activity in accordance with strategy prevalence are present across distinct global contexts that cause large-scale restructuring of the ensemble activity (see previous point). We therefore favor retaining the original wording.

(vi) tags ACC representations with contextual content = ACC encodes context?

Our goal was to highlight that ACC representation of *individual strategies* is modified in distinct contexts. Thus, both the identity of the specific strategy and the global behavioral context affect ACC neural activity – something that would not be captured by saying “ACC encodes context”. We do appreciate, however, that this would be somewhat incomprehensible in the introduction; that part is now greatly simplified in the revision.

(vii) organizing principles for the ACC ensemble dynamics = how ACC encodes information?

Our goal was to set up the contrast between the marked structuring of the ACC dynamics we observe in this stud (Figure 5) and the emerging picture from past findings that there is seemingly little task-related activity at the level of single units. We suspect it is less about how ACC encodes information and more about what information it encodes. We have reworded the offending bit to state

“raising the possibility that more structured responses in the ACC remain to be discovered.”

(viii) representations of individual strategies are also marked with contextual content = encoding of strategy per se?

As we detailed in our answer to suggestion (vi), the fact that two separate effects contribute to shaping ACC neural activity leads us to favor our more comprehensive, if somewhat more cumbersome, wording over one that highlights either context encoding or strategy encoding alone.

Results and a more general description of the analytical methods should be in the main text rather than the methods. Analytical methods are hard to follow in some places.

We have expanded the discussion of our approach to establishing strategy encoding in the ACC, as well as elaborated on our approaches to eliminating the major confounds. We hope that the Reviewers will find that the revised version strikes a better balance between providing enough details while still presenting a smooth narrative.

**A more incremental and linear exposition would help readers understand the data and analyses.**

We have expanded the following arguments:

– Deviations from dominant sequence targets are not mere errors of execution

– Action encoding in the ACC is strategy specific

– Large-scale re-organizations of ACC ensemble do not reflect a mere ‘reward’/’no reward’ dichotomy

– Motor parameters cannot explain observed activity modulation

We have also introduced a separate, strongly elaborated section on the contribution of reward encoding.

We hope that the Reviewers will deem the new narrative easier to follow for the reader.

**Point-by-point Responses to Other Reviewer Comments**

Reviewer 1“In analyzing neural activity accompanying the behavioral persistence of the dominant sequence after a block change, the authors find that the ACC ensemble firing pattern is closer to the original dominant sequence pattern during reinforcement and less like this pattern during exploration… As time, and trials, progress the rat is approaching the point at which it explores another strategy**.** The authors find strengthened "prevalence" encoding with increasing sequence repetition, but if this parameter is related to behavioral change/flexibility, this was not clear to me. Might there be something unique about the last trials in a tail "predicting" an upcoming switch? Can the authors please expand? Relatedly, if the prediction of upcoming behavioral change is not observed in the neural activity from sequence steps 2-6, it is notable that these are the steps 'within' the sequence, that leaves out the initiation (first center poke) and termination (reward/reward omission). Thus one could imagine this information is "missed" in the current analysis given that both the reward period and the initiation of a trial at the center are not analyzed. This does lead me to suggest a softening of some claims made of identifying "unifying principles" of ACC function, as the authors state, based on the analyses included in the current report, since the neural activity related to the full unit of behavior is not considered. (I appreciate the motivation behind this focus on within-sequence behavior – the wish to compare time periods with similar movement parameters.)

We apologize for the confusion; while the sequence prevalence itself tends to be high for ‘dominant tails’, we do not claim that the fit of the prevalence model is *better* at those sequence instances. We do share the interest in linking prevalence encoding to behavioral adaptation as well as the Reviewer’s intuition that block transitions should be among the epochs where strategy prevalence is tracked particularly well. And indeed, we had spent a considerable amount of time thinking about whether we can identify and interpret periods during the session where our prevalence model fits better or worse. Two arguments convinced us to abandon that direction: a technical one and a conceptual one. The technical argument is that when the explanatory power of a variable is limited, regression residuals are proportional to the variable itself. Thus, any meaningful comparison of the model’s fit would have had to be done for periods where strategy prevalence is within a similar range. The conceptual argument is even more disarming: imagine we do identify a putative session epoch where the model fits worse. While it is possible that it truly means that the animal tracks the details of how much he has pursued this strategy in recent past less, it is equally possible that we were simply off in selecting the specific window over which the prevalence signal is estimated, the exact behavioral statistic tracked, or the exact form of the dependence between that statistic and neural activity. We certainly do see changes leading up to behavioral switches at block transitions – something we plan to elaborate on in a subsequent paper – but whether those are related to prevalence tracking is something we believe is hard to crack.

We have now made sure that we don’t make the claim that the structuring of activity by prevalence during strategy execution is a ‘unifying principle’ either the Results or the Discussion sections.

Reviewer 2

(3) … (iii) ACC neurons might code an action sequence as a whole or configuration, much like a compound action, which might instead be interpreted as the authors' specific content of strategy prevalence. However, further evidence is needed to rule out the possibility that neural activities are merely related to a complex action per se rather than a higher cognitive cause. While it is understandable that confounds are not easy to control fully, the authors should emphasize these limitations more in the discussions.

We apologize for not making the foundational premise clearer: all comparisons of activity dynamics in this manuscript are always done for different instances *of the same multistep sequence*. As such, the modulation by sequence prevalence we describe here is independent of the specific form in which a multi-step strategy is encoded in the first place. It is precisely the type of concerns insightfully expressed here by several Reviewers that convinced us that ‘across-instances-of-same-sequence’ are the only interpretable comparisons that can be performed.

(4) Recording data show that neural correlates of strategy prevalence in M2 are slightly higher than in SMC, but inactivating SMC but not M2 affects corresponding behavior. How do the authors explain the disagreement between recording and inactivation data?

We believe that the role that SMC plays in this behavior does not involve tracking the statistics of chosen strategies in recent past.

More importantly, behavioral data on the inactivation of ACC is missing, although cannula implantation surgery in ACC is mentioned in the manuscript. More clarifications are needed for these questions.

The behavioral effect of ACC inactivation has been reported in our previous publication (see Figures 3 and S2 in (Tervo et al., 2014)).

Reviewer 3

3) I am not convinced that it is valid to compare the 3 sequence responses to single responses in the 'competitor task' (Figure 1.d). The reward expectation is much different because there is a non-zero probability of reward after every choice. Furthermore, readers need to know much more information. Are these different animals with distinct training regiments, differences in the apparatus, etc.? Presumably, rats are going to the feeder after responses in the competitor task more often than the sequence task. I did not find in the present manuscript where the feeder was located relative to the response ports.

We apologize for the several sources of confusion here! First, we compared a sequence of the same 3 responses in both conditions, i.e. a ‘LLR’ sequence in both the competitor and the sequence tasks. Second, in all our tasks – done in the same behavioral box – liquid reward is delivered directly at the choice ports. We have now included that information in the figure legend. That said, we appreciate that this was not the ideal comparison for strategy encoding. We hope that the Reviewer will find that our newly added analyses (see section ‘Strategy-specific action encoding’ in Essential Revisions above) strengthen the main claim of that section to their satisfaction.

4) My primary critique of the paper is that alternate explanations for variance are not sufficiently tested. Presently, this is done very piecemeal and is scattered throughout the results.

We have now expanded a number of these arguments, and, most importantly, consolidated the discussion of the reward contribution into its own section. Please refer to the section *‘Activity modulation by strategy prevalence vs by reward or certainty’* in Essential Revisions above for more details.

The authors do address some of the possible confounders, but not to a level of sufficient rigor to support the overall claim that it is a dominant/exploratory strategy that accounts for differences in neural firing. In no particular order, I think the primary confounders are:reward expectancy;

Reward expectancy is a confusing concept; it may mean different things to different readers. At one level, all behavioral choices may be framed as being associated with reward expectancy, even when there are no specific reward expectations. The concept is even further confounding because of the inadequate specification of reward during many discussions—social approbation and social stimuli, relief from anticipated costs, and even knowledge may be as rewarding outcomes as appetitive consumables. Indeed, even hypothesis falsification presumably rarely centers on completely implausible choices and almost always yields clarifying information.

Given this tight conceptual coupling between sought outcomes and agent’s choices, at one extreme, one could argue that reward expectancy is reflected in the extent to which an agent devotes attention to any specific strategy – an interpretation that also relates to the concept of subjective value. We would argue that this interpretation of reward expectancy is semantically equivalent to what we capture here with the local strategy prevalence. However, reward expectancy can also apply to any singular action, or specific strategy. We tackle these concepts now in a new section of the manuscript titled “The tracking of reward by the ACC neural ensemble is strategy-specific, however reward prevalence is insufficient to account for activity modulation.” And while we do not claim to have explicit knowledge of the specific updating rule used by each animal for its reward expectation, we use multiple linear regression analysis to determine whether the unexpected robustness of the simple strategy prevalence model in explaining a marked fraction of ACC neural activity variance during the execution of a specific sequence derives mostly from the impact of successfully procured reward. We believe that the proper approach to the latter question requires us to apply the same updating rule to the reward term as we did to the prevalence one. Please refer to the section ‘Activity modulation by strategy prevalence vs by reward or certainty’ in Essential Revisions above for more details.

­

response vigor;

We apologize that our piecemeal presentation in the original submission made this part difficult to find; we used execution time as a way to capture response vigor. The expanded analysis of this parameter, along with movement trajectory, is now detailed in the following section (p 16):

“The comparatively low amount of neural activity variance in the motor regions M2 and SMC explained by strategy prevalence suggests that the observed modulation is unlikely to reflect movement parameters that may co-vary with changes in the animal’s dedication to a particular strategy. To establish this more explicitly, we directly quantified trial-by-trial measures of movement vigor (sequence execution time) and kinematics (the first principal component of movement trajectory) for each instance of sequence execution and then asked how well linear models that incorporate these parameters performed at explaining variance in ACC firing rates (Methods). We first evaluated the performance of an equivalent linear model that used either execution time or the first principal component of movement trajectory as a single parameter; neither model could explain ACC neural activity variance as well as sequence prevalence (Figure 4 —figure supplement 3a). We then focused in more detail on the dataset recorded in the anterior part of the cingulate cortex that displayed the strongest modulation by strategy prevalence (Figure 4b,c), and asked whether adding a separate movement-related parameter to the linear model would dwarf the explanatory power of the prevalence term (Methods). To ensure that the resulting weights on the two parameters would directly reflect their relative contributions, we z-score normalized all the variables to 0 mean and standard deviation of 1 prior to fitting the model. Consistent with the relatively poor performance of the single movement-related parameter models, the expanded models revealed only a minor contribution from those motor parameters to the overall model’s performance (Figure 4 —figure supplement 3b). While it remains possible that more nuanced aspects of movement, like the animal’s posture, contribute to the observed activity modulation, these observations suggest that gross movement-related parameters are not behind the explanatory power of strategy prevalence. “

posture during nose poke (which may depend on prior choice), position in sequence; side chosen,

*We apologize for the confusion.* With the exception of the strategy decoding section, all of the neural data analyses in the manuscript compare representations of the same given action sequence, just across different instances of its execution. Furthermore, the time window around the center port entry on the first step is excluded from the analyses to minimize the contribution from the previous sequence. Thus, while small variations in posture are still possible, those dependent on history, position in sequence or side chosen would be the same across different instances of the same action sequence, and thus would not confound our analyses.

Nevertheless, to highlight the fact that the resolution of our video data precludes us from doing detailed posture tracking, we have rounded up the section on motor confounds with the following statement:

“While it remains possible that more nuanced aspects of movement, like the animal’s posture, contribute to the observed activity modulation, these observations suggest that gross movement-related parameters are not behind the explanatory power of strategy prevalence. “

a relative time during the session (motivation and/or typical run-down of neural firing during long operant tasks),

Figure 3c-d presents the easiest way to see that time during the session cannot explain the observed effects. Indeed, the two blocks of ‘dominant’ ‘RRL’s in the example in 3c, and more generally across the dataset, are more distant from each other in time than each pairing of ‘dominant’ vs ‘exploratory’ blocks. Yet, the representations across the two dominant blocks are always more similar than those between ‘dominant’ and ‘exploratory’.

and rule certainty.

We have now expanded the set of contexts where a given behavioral sequence was performed under a ‘no-reward’ condition. Specifically, we have now collected new data in a setting where persistent, but unrewarded ‘LLR’ and ‘RRL’ sequence instances dominated within the unfamiliar ‘LLLR’ and ‘RRRL’ contexts. What makes this fundamentally different from the case of ‘dominant tails’ is that the observed persistence with an unrewarded sequence in this case occurred after the animals had detected an unsignalled block change and switched away from the previous strategy and thus cannot be explained by a lack of rule switch awareness.

The addition of this extra 'no reward’ condition permitted us to more convincingly demonstrate (a) that the global ‘dominant’ ACC ensemble state relates to strategy dominance rather than the associated reward, and (b) that strategy prevalence makes at least some reward independent contribution to modulating ACC activity during the execution of a specific strategy. Critically, since rule certainly captures one’s belief about strategy outcome, observing activity modulation by strategy prevalence under conditions of long-term persistence with an unrewarded strategy argues against that modulation reflecting rule certainty.

Neural activity often drifts over the course of long sessions in which animals perform many trials of an operant task. The authors need to dismiss that this is a possibility, particularly in their analysis of dominant vs exploratory strategy. It is not evident from the manuscript how many trials animals are doing and how long the blocks are. Because of the number of possible sequences, it seems likely that the time between each strategy being the dominant versus exploratory may allow for task-independent drift (fatigue or mechanical instability) in neural activity. Authors should show that difference in firing for dominant/exploratory strategy is not influenced by drift in neural firing over time. If it is, this variance must be parsed out.

Figure 3c-d presents the easiest way to see that activity drift during the session cannot explain the observed effects. Indeed, the two blocks of ‘dominant’ ‘RRL’s in the example in 3c, and more generally across the dataset, are more distant from each other in time than each pairing of ‘dominant’ vs ‘exploratory’ blocks. Yet, the representations across the two dominant blocks are always more similar than those between ‘dominant’ and ‘exploratory’.

We write on p. 11:

“Moreover, the mean Euclidean distance in the activity state space between two ‘dominant’ blocks separated in time was significantly smaller than the mean pairwise distances between a dominant and exploratory block, arguing against the possibility that the observed representational transitions in ACC arose from an instability in neural recordings (Figure 3 c,d).”

6) I recommend tempering the language in the abstract, which mentions intelligence and complex settings. The relation of the present work to these topics is better presented in the discussion if the authors think them relevant.

We have now removed any reference to intelligence from the abstract.

7) The firing rates shown in Figure 1 and 4 appear quite high for rat pyramidal neurons in mPFC. Are these cells typical of the dataset, or outliers in terms of firing rate? A quantification of the general activity over the population of cells is needed. Moreover, a plot of firing rate versus spike width would provide valuable insight into the distribution of putative pyramidal and interneurons.

It is indeed typical to see *peak firing rates* of 20-60 Hz in rat mPFC neurons around behaviorally-relevant events (see, for example, Figure 3B in (Tang et al., 2023) for a navigation task, and Figure 3B in (Murakami et al., 2017) for a decision-making task). The average firing rates for those neurons are much lower (see Author response image 1). Unfortunately, unlike for hippocampal neurons, spike width is not considered a robust criterion for cleanly separating cortical pyramidal cells and interneurons. Nevertheless, the preponderance of relatively low-firing (on average) cells in our dataset, and the wide-spread nature of both the global and the local modulation at the core of this manuscript suggests that pyramidal cells play a prominent role in the described encoding.

**Author response image 1. sa2fig1:** 

Boorman, E.D., Behrens, T.E., Woolrich, M.W., and Rushworth, M.F. (2009). How green is the grass on the other side? Frontopolar cortex and the evidence in favor of alternative courses of action. Neuron 62, 733-743.Hayden, B.Y., and Platt, M.L. (2010). Neurons in anterior cingulate cortex multiplex information about reward and action. J Neurosci 30, 3339-3346.

Kennerley, S.W., Dahmubed, A.F., Lara, A.H., and Wallis, J.D. (2009). Neurons in the frontal lobe encode the value of multiple decision variables. J Cogn Neurosci 21, 1162-1178.

Kolling, N., Behrens, T.E., Mars, R.B., and Rushworth, M.F. (2012). Neural mechanisms of foraging. Science 336, 95-98.

Luk, C.-H., and Wallis, J.D. (2013). Choice coding in frontal cortex during stimulus-guided or action-guided decision-making. Journal of Neuroscience 33, 1864-1871.

Murakami, M., Shteingart, H., Loewenstein, Y., and Mainen, Z.F. (2017). Distinct sources of deterministic and stochastic components of action timing decisions in rodent frontal cortex. Neuron 94, 908-919. e907.

Shidara, M., and Richmond, B.J. (2002). Anterior cingulate: single neuronal signals related to degree of reward expectancy. Science 296, 1709-1711.

Tang, W., Shin, J.D., and Jadhav, S.P. (2023). Geometric transformation of cognitive maps for generalization across hippocampal-prefrontal circuits. Cell reports 42.

Tervo, D.G., Proskurin, M., Manakov, M., Kabra, M., Vollmer, A., Branson, K., and Karpova, A.Y. (2014). Behavioral variability through stochastic choice and its gating by anterior cingulate cortex. Cell 159, 21-32.